# Two consecutive microtubule-based epithelial seaming events mediate dorsal closure in the scuttle fly *Megaselia abdita*

Juan Jose Fraire-Zamora[1,2]*, Johannes Jaeger[2,3,4†], Jérôme Solon[1,2]*

[1]Cell and Developmental Biology Programme, Centre for Genomic Regulation (CRG), The Barcelona Institute of Science and Technology, Barcelona, Spain; [2]Universitat Pompeu Fabra, Barcelona, Spain; [3]System Biology Programme, Centre for Genomic Regulation (CRG), The Barcelona Institute of Science and Technology, Barcelona, Spain; [4]Konrad Lorenz Institute for Evolution and Cognition Research (KLI), Klosterneuburg, Austria

**Abstract** Evolution of morphogenesis is generally associated with changes in genetic regulation. Here, we report evidence indicating that dorsal closure, a conserved morphogenetic process in dipterans, evolved as the consequence of rearrangements in epithelial organization rather than signaling regulation. In *Drosophila melanogaster*, dorsal closure consists of a two-tissue system where the contraction of extraembryonic amnioserosa and a JNK/Dpp-dependent epidermal actomyosin cable result in microtubule-dependent seaming of the epidermis. We find that dorsal closure in *Megaselia abdita,* a three-tissue system comprising serosa, amnion and epidermis, differs in morphogenetic rearrangements despite conservation of JNK/Dpp signaling. In addition to an actomyosin cable, *M. abdita* dorsal closure is driven by the rupture and contraction of the serosa and the consecutive microtubule-dependent seaming of amnion and epidermis. Our study indicates that the evolutionary transition to a reduced system of dorsal closure involves simplification of the seaming process without changing the signaling pathways of closure progression.
DOI: https://doi.org/10.7554/eLife.33807.001

*For correspondence:
juanjose.fraire@crg.eu (JJF-Z);
jerome.solon@crg.eu (JS)

Present address: †Complexity
Science Hub (CSH), Vienna,
Austria

**Competing interests:** The
authors declare that no
competing interests exist.

**Reviewing editor:** Maithreyi
Narasimha, Tata Institute of
Fundamental Research

## Introduction

Mechanical forces produced at the cellular level are known to shape tissues during morphogenesis (see *Lecuit et al., 2011*, for a recent review). Molecular motors and cytoskeletal elements generate these mechanical forces, which cause tissues to deform and change shape (*Mammoto and Ingber, 2010*). Until recently, such tissue-level aspects of morphogenesis have received relatively little attention in the field of evolutionary developmental biology. The evolution of developmental processes is generally attributed to changes in genetic regulation (see for example, *Carroll et al., 2009*; *Davidson and Erwin, 2006*; *Peter and Davidson, 2015*; *Wilkins, 2002*). To date, it is not fully understood how a developing organism integrates the mechanical and genetic factors necessary to shape a tissue, or how this interplay between tissue mechanics and genetics is contributing to the evolution of development. We focus on this latter aspect by studying how a continuous epidermal layer is formed by epithelial fusion during dorsal closure in a non-model organism, the scuttle fly *Megaselia abdita* (Diptera: Phoridae).

Epithelial fusion is a fundamental morphogenetic mechanism in animal development where two opposing epithelial sheets are brought together to subsequently seam and result in a single continuous epithelial layer (*Jacinto et al., 2001*). Dorsal closure in *Drosophila melanogaster* (Diptera: Drosophilidae) is a classical model system to study epithelial fusion (*Jacinto et al., 2000*). This process is promoted by the mechanical action of different players: a contractile actomyosin cable forming at

**eLife digest** For a single fertilized egg to become an animal with many millions of cells, complex networks of genes must control the different stages of development. These gene networks create all the patterns needed to form different parts of the body. Changes to these patterns can create new species, with different sizes, body shapes, colors and lifestyles.

Researchers often examine how evolution can create new species by altering gene networks to change patterns of development. Yet, some differences in development may not directly result from changes to gene networks. Other causes could include how the tissues are organized to begin with. One way to better understand this kind of difference is to compare developmental processes between two or more related species.

Dorsal closure, for example, is a stage in a fly's development when one layer of cells – the epidermis – closes over the back of the developing embryo. Dorsal closure in the scuttle fly (*Megaselia abdita*) involves the epidermis and two other layers of cells, yet it only involves one other cell layer in the vinegar fly (*Drosophila melanogaster*). The different number of layers means that dorsal closure must happen differently in the two species. Yet, Fraire-Zamora et al. now report that the genetic control behind the process is very similar in both species. Instead, it is differences in the arrangement and shape of cell layers that lead to the changes in dorsal closure.

Dorsal closure involves physical changes to several cell layers, and is driven by protein structures that give shape and strength to cells. These findings highlight how living systems adapt to evolutionary changes. Fraire-Zamora et al. suggest that the same concepts could be adapted further, helping to design and build complex organs in the laboratory. This in turn could help scientist develop new approaches to repairing tissues and healing wounds.

DOI: https://doi.org/10.7554/eLife.33807.002

the leading edge of the epidermal flanks, the extraembryonic amnioserosa which covers the dorsal opening and generates contractile forces during epidermal flank advancement, and the eventual seaming of the epidermis through a mechanism involving microtubule-based cellular protrusions (*Eltsov et al., 2015*; *Hutson et al., 2003*; *Kiehart et al., 2000*; *Saias et al., 2015*). Genetically, the c-Jun N-terminal kinase (JNK) pathway and the transforming growth factor beta (TGF-β) family gene *decapentaplegic* (*dpp*) play an essential regulatory role in the process (*Fernández et al., 2007*; *Glise and Noselli, 1997*; *Jacinto et al., 2002*; *Knust, 1997*). The expression of *dpp* localizes to the leading edge of the epidermal flanks and depends on the activity of the *D. melanogaster* JNK gene (*basket*, *bsk*). Embryos lacking *bsk* activity show downregulation of *dpp* at the epidermal leading edge, failure of dorsal closure progression, and a dorsal-open phenotype in the larval cuticle (*Glise and Noselli, 1997*; *Sluss et al., 1996*). At the molecular level, activation of the JNK/Dpp signaling pathways promotes the formation and maintenance of the actomyosin cable at the epidermal leading edge (*Ducuing et al., 2015*) and, thus, progression of the opposing epidermal flanks toward the dorsal midline where they meet. At the final stage of dorsal closure, the opposing epidermal flanks 'zipper' or 'seam' through the action of microtubules that align toward the dorsal opening and promote the formation of filopodial protrusions at both epidermal leading edges (*Jacinto et al., 2002*; *Jankovics and Brunner, 2006*; *Millard and Martin, 2008*).

Dorsal closure is a conserved morphogenetic process that occurs in all insects (*Chapman, 1998*). Although in *D. melanogaster* it involves two tissues, the embryonic epidermis and the extraembryonic amnioserosa, in most insects it involves three: the embryonic epidermis, an extraembryonic amnion, and a separate extraembryonic serosa (*Panfilio, 2008*; *Schmidt-Ott and Kwan, 2016*). These complex anatomical differences raise the question whether the mechanisms responsible for epithelial fusion in a simple two-tissue system are conserved in a three-tissue system. The phorid scuttle fly *M. abdita* (placed in an early branching cyclorraphan lineage) presents a three-tissue system of dorsal closure and has been established as a model to study the evolution of developmental processes (*Bullock et al., 2004*; *Rafiqi et al., 2008*; *Schmidt-Ott et al., 1994*; *Stauber et al., 2000*; *Wotton et al., 2015*). Thus, *M. adbita* offers the opportunity to compare the three-tissue system of dorsal closure to the two-tissue system present in *D. melanogaster*.

Here, we perform a quantitative characterization of dorsal closure in *M. abdita*. Combining molecular tools with live imaging, we show that dorsal closure in *M. abdita* embryos occurs in three distinct phases: (i) serosa rupture and retraction, (ii) serosa contraction and progression of opposing epidermal flanks, and (iii) a dual seaming process to eventually form a fused continuous epidermis. Despite the significant morphological differences with *D. melanogaster*, the regulation of dorsal closure in *M. abdita* involves a conserved role for the JNK/Dpp signaling pathway to form and maintain an epidermal actomyosin cable surrounding the dorsal opening. More specifically, we find that following an actomyosin-dependent contraction of the serosa, two consecutive microtubule-dependent seaming events take place in the amnion as well as in the epidermis. In both cases, apical microtubule bundles align and extend toward the site of closure suggesting a general epithelial fusion mechanism. Altogether, our results provide a dynamic and quantitative description of epithelial fusion in a complex three-tissue system. They indicate that the evolutionary transition from a three-tissue to a two-tissue system of dorsal closure involves changes in the number and sequence of morphogenetic events, rather than changes in the spatio-temporal activity of the main signaling pathways that control closure progression.

## Results

### Dorsal closure in *Megaselia abdita* involves synchronized serosa rupture and epidermal progression

In order to map the spatial arrangement of tissues involved in dorsal closure of *M. abdita* embryos, we obtained confocal projections of fixed non-devitellinized embryos with stained nuclei. Nuclear anatomy and staining have been used previously to identify extraembryonic tissues in the flour beetle *Tribolium castaneum* (*Panfilio et al., 2013*). In *M. abdita*, staining fixed embryos with the nuclear dye DAPI allowed us to distinguish three types of tissues: (*1*) The extraembryonic serosa, which constitutes the outermost extraembryonic layer and envelops the entire *M. abdita* embryo before the onset of dorsal closure (magenta in *Figure 1A, A' and and B,B'*). Its cells have very large nuclei (average size $125 \pm 21$ µm², SD, *n* = 150 cells) and show discontinuous or 'punctuated' DAPI staining (magenta in *Figure 1A and A'* and *Figure 1—figure supplement 1A–A' and B–B''*). (*2*) The extraembryonic amnion, which is one to two cells wide, localizes in between the serosal and epidermal tissues (blue in *Figure 1A, A' and and B,B'*). Its cells also have large nuclei (average size $77 \pm 16$ µm², SD, *n* = 150 cells) and show a more continuous, 'compact' DAPI staining (blue in *Figure 1A and A'* and *Figure 1—figure supplement 1A–A' and B–B''*). (*3*) The embryonic epidermis, which contains numerous small nuclei (average size $14 \pm 3$ µm², SD, *n* = 150 cells) that are tightly packed (gray in *Figure 1A and A'*).

In order to obtain a closer view of the spatial arrangement of tissues in live *M. abdita* embryos, we injected DAPI at the embryo poles during dorsal closure stage and obtained confocal projections. This staining showed that amnion cells sit on top of yolk granules and are positioned adjacent to the embryonic epidermis (blue arrowheads in *Figure 1—figure supplement 1C–C''*).

When fixing *M. abdita* embryos at dorsal closure stage, devitellinization also removes the serosa together with the vitelline membrane (*Figure 1—figure supplement 2A*). Devitellinization and serosa cells removal resulted in a gap on the dorsal side of the embryo, seen as lack of phalloidin staining (*Figure 1—figure supplement 2B*). Amnion cells (1–2 cell rows adjacent to the epidermis, blue arrowheads in *Figure 1—figure supplement 2B*) remained apposed to an intact epidermis. In a few cases, devitellinization left some intact serosa cells on top of *M. abdita* embryos (very large cells highlighted by phalloidin and DAPI counterstains, white arrowhead in *Figure 1—figure supplement 2C and C'*). An optical re-slice of confocal projections of the intact serosa and amnion cells showed that these two cells types are apposed (yellow arrowhead in *Figure 1—figure supplement 2C''*).

In summary, the anatomy of *M. abdita* embryos at dorsal closure reveals a three-tissue system, where large serosa cells surround the embryo and are apposed to the amnion cells at the dorsalmost end of the embryo. Amnion cells form a row, in turn apposed to the adjacent epidermis (*Figure 1B and B'*). This three-tissue geometry poses an interesting challenge for the process of dorsal closure, since the apposed serosa and amnion need to undergo dramatic rearrangements to achieve epidermal fusion at the end of dorsal closure.

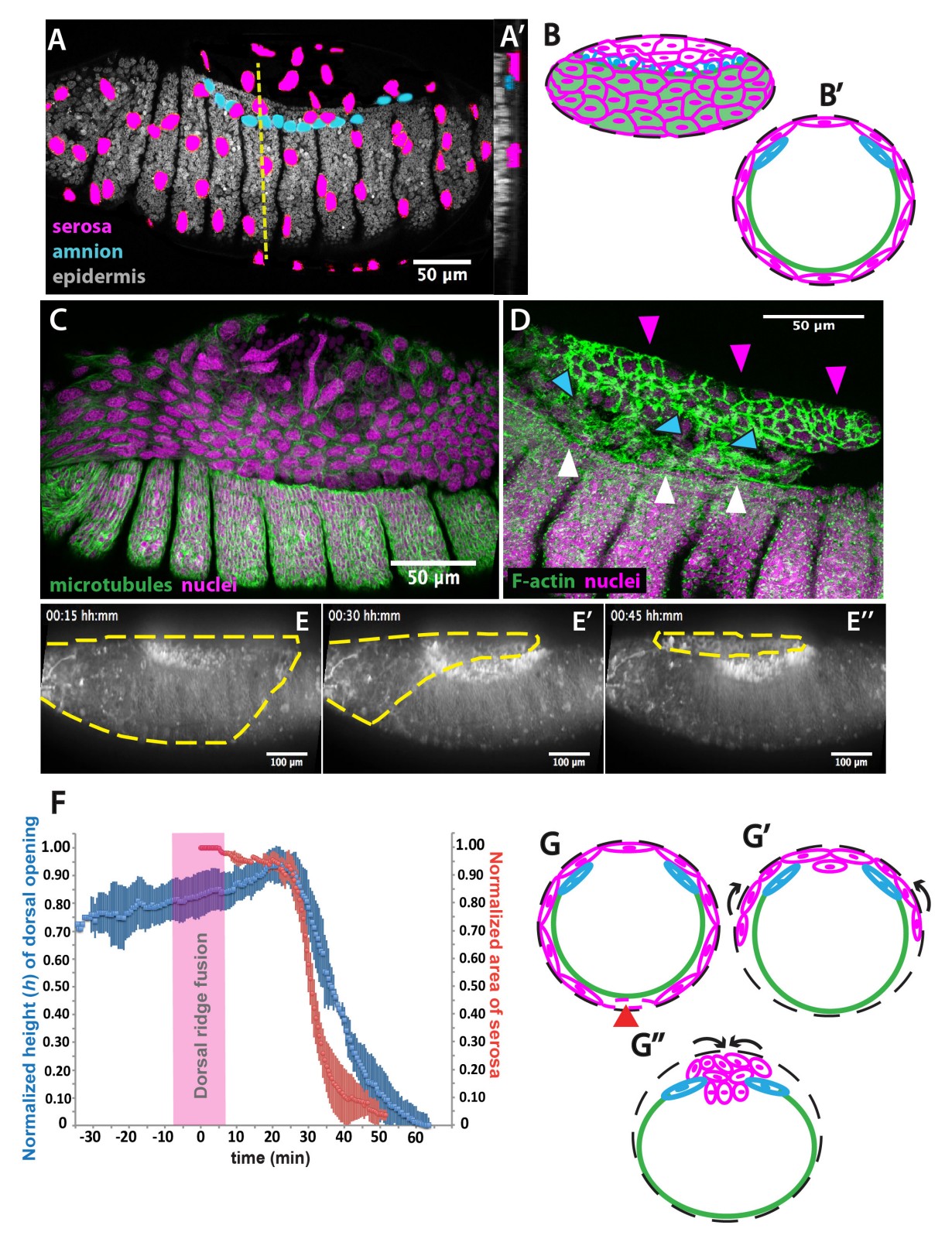

**Figure 1.** The extraembryonic serosa ruptures and accumulates dorsally previous to epidermal seaming in *Megaselia abdita*. (A) Nuclear staining of *M. abdita* embryos prior to dorsal closure reveals three types of tissues: the extraembryonic serosa (magenta), the extraembryonic amnion (blue), and the embryonic epidermis (gray). (A') An orthogonal re-sliced stack along the dashed yellow line in A shows the position of the embryonic (gray), amniotic (blue), and serosal (magenta) tissues in transverse view. (B) Schematics depicting the organization of the serosa cells (magenta), amnion cells (blue), and

*Figure 1 continued on next page*

*Figure 1 continued*

embryonic epidermis (green) in lateral and (**B'**) transverse view. The black dashed line represents the vitelline envelope (**C**) *M. abdita* embryo undergoing rupture and retraction of serosal tissue along the ventral side. Staining against β-tubulin in green, and DAPI nuclear counterstain in magenta. (**D**) Serosal cells (magenta arrowheads) accumulate on the dorsal side of the embryo after rupture. The serosa remains apposed to the amnion (blue arrowheads), which is in turn apposed to the embryonic epidermis (white arrowheads). Phalloidin stain in green and DAPI nuclear counterstain in magenta. (**E**) Images from a time-lapse sequence of serosa retraction in a *M. abdita* embryo injected with the fluorescent lipophilic dye FM 4–64 (from *Video 1*). Yellow dashed line shows the contour of the serosa covering the embryo during retraction. (**F**) Relative changes in area of the serosa during retraction (red, *n* = 15 embryos), and relative changes in height (*h*) of the dorsal opening (blue, *n* = 15 embryos) during dorsal closure in *M. abdita*. Vertical bars represent standard deviation (SD). Time range of dorsal ridge fusion is represented by pink area (*n* = 15 embryos) as a landmark for the initiation of dorsal closure. The origin of the time axis (*T = 0 min*) is set at the point of serosa rupture. (**G**) Schematics depict transverse views of embryos during serosa rupture, retraction, and dorsal accumulation. Serosal cells rupture along the ventral end of the embryo (red arrowhead). (**G'**) The remaining lateral serosa cells retract towards the dorsal side (black arrows). (**G''**) Serosa cells continue retracting until they completely accumulate onto the dorsal opening (black arrows). Color scheme as in B. In all embryos and schematics, dorsal is to the top. Embryos in A, C, D and E show lateral views where anterior is to the left.

DOI: https://doi.org/10.7554/eLife.33807.003

The following figure supplements are available for figure 1:

**Figure supplement 1.** Identification of extraembryonic tissues in *Megaselia abdita*.
DOI: https://doi.org/10.7554/eLife.33807.004
**Figure supplement 2.** Description of the three-tissue system anatomy in *Megaselia abdita* prior to dorsal closure.
DOI: https://doi.org/10.7554/eLife.33807.005
**Figure supplement 3.** Anatomical landmarks during dorsal closure progression and serosa internalization.
DOI: https://doi.org/10.7554/eLife.33807.006
**Figure supplement 4.** Tissue topology at anterio-posterior positions of different stages of dorsal closure in fixed embryos of *Megaselia abdita*.
DOI: https://doi.org/10.7554/eLife.33807.007

How does *M. abdita* solve this problem? At early stages of dorsal closure, the serosa surrounding the embryo undergoes an abrupt rupture (*Wotton et al., 2014*). Serosa rupture initiates close to the posterior pole of the embryo. Spread of the rupture occurs anteriorly through the ventral side of the embryo and results in a retraction of the serosa toward the dorsal end of the embryo (*Figure 1C*). During this process, serosa cells accumulate at the dorsal opening (magenta arrowheads in *Figure 1D*, and magenta cells in 1 G-G''). In the meantime, the amnion remains in place, apposed to the serosa (blue arrowheads in *Figure 1D*, and blue cells in 1 G-G''), and adjacent to the embryonic epidermis (white arrowheads in *Figure 1D*, and green line in 1 G-G'').

To gain quantitative evidence on the dynamics of dorsal closure in *M. abdita*, we labeled live embryos with the fluorescent lipophylic dye FM 4–64 and followed the process using confocal imaging (*Video 1*). We used the fusion of the dorsal ridge (merging of the ridge primordia at the dorsal midline, magenta bar in *Figure 1—figure supplement 3A and A'*) as a developmental landmark for the initiation of dorsal closure (*T = 0 min*) (*Campos-Ortega and Hartenstein, 1997*; *VanHook and Letsou, 2008*). Under this time frame, the serosa ruptures at *T = 25 min* ($\pm 8$ min, SD; *n* = 15 embryos) and dorsal closure (seaming of the embryonic epidermal flanks at the dorsal midline, see yellow line in *Figure 1—figure supplement 3A''*) concludes at *T = 74 min* ($\pm 10$ min, SD, *n* = 15 embryos).

To relate serosa retraction to the kinetics of dorsal closure, we measured the relative changes in area of the serosa covering the embryo and the relative changes in height (*h*) of the dorsal opening over time (see red and blue curves in *Figure 1F*, yellow dashed line in *Figure 1E*, blue line in *Figure 1—figure supplement 3A*- and Materials and methods). At *early* stages of dorsal closure, the leading edge of the epidermis straightens (yellow line in *Figure 1—figure supplement 3A and A''*) and the dorsal opening increases progressively in height (*h*) (blue curve in *Figure 1F*). After dorsal ridge fusion (pink area in *Figure 1F*), the serosa ruptures, retracts and accumulates onto the dorsal opening (*Figure 1C*, magenta arrowheads in 1D, yellow dashed line in 1E, magenta cells in 1G'-G''). This is concurrent with a fast reduction in dorsal opening height (red and blue curves in *Figure 1F*, yellow line in *Figure 1—figure supplement 3A and A''*). Visualizing the process from an orthogonal view, we observe that following serosa accumulation on top of the dorsal opening, this extraembryonic tissue bends inwards and serosa cells undergo an apicobasal elongation resulting in the internalization of a large part of the serosa cells into the yolk prior to epidermal fusion (*Video 2* and *Figure 1—figure supplement 3B and C–C'*). Optical re-slices of confocal projections along the

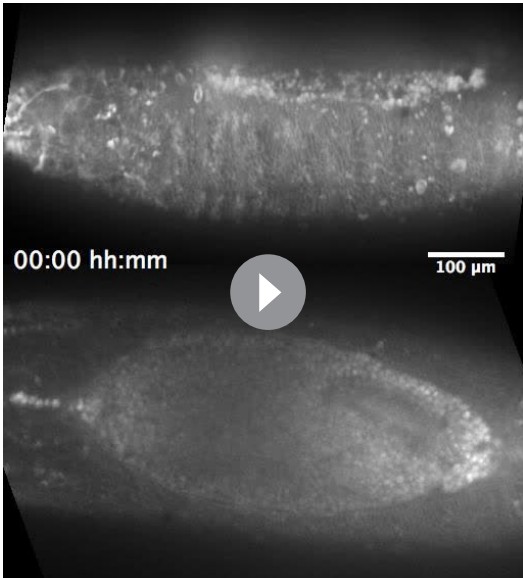

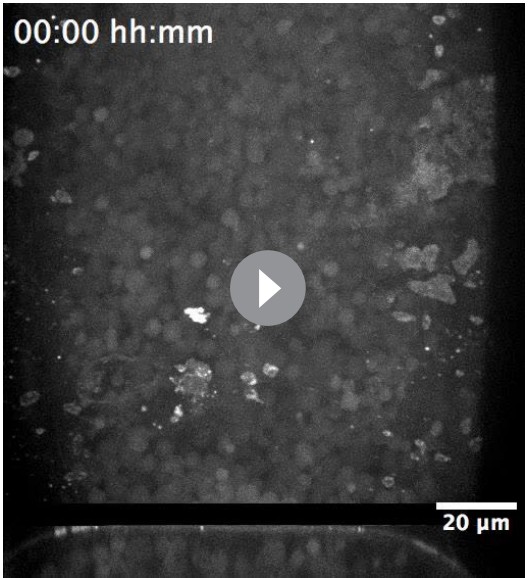

**Video 1.** Dorsal closure in *Megaselia abdita* involves the rupture and retraction of the serosa and advancement of the epidermal flanks to the dorsal midline. Time-lapse sequence of dorsal closure in two *M. abdita* embryos injected with the fluorescent label FM 4–64. Top: serosa rupture and retraction in lateral view. Rupture initiates at a ventral-posterior location and spreads anteriorly along the ventral side of the embryo. The ruptured serosa accumulates on the dorsal side, where epidermal seaming occurs at the end of dorsal closure. Bottom: dorsal view of the initiation of dorsal closure, marked by the fusion of the dorsal ridge and straightening of the epidermal leading edge (see *Figure 1—figure supplement 3A*). Epidermal flanks are brought together to the dorsal midline where epidermal seaming occurs. In both embryos, anterior is to the left.
DOI: https://doi.org/10.7554/eLife.33807.008

**Video 2.** The extraembryonic serosa of *Megaselia abdita* embryos internalizes into the yolk prior to epidermal seaming. Time-lapse sequence of a dorsal (top) and orthogonal view (bottom) of an FM 4–64-labeled embryo. After serosa rupture and accumulation at the dorsal opening, the extraembryonic tissue internalizes into the yolk, as observed by an inward bending of the tissue and apicobasal cell elongation. Upon internalization of the serosa, the epidermal flanks advance and fuse at the dorsal midline. In the dorsal view, anterior is to the top. In the orthogonal view, dorsal is to the top.
DOI: https://doi.org/10.7554/eLife.33807.009

anterio-posterior axis of fixed embryos show that extraembryonic tissues internalizes during dorsal closure and do not accumulate underneath the fused epidermis after completion of the process (*Figure 1—figure supplement 4*). Thus, at *late* dorsal closure stages, the serosa is fully internalized (magenta cells in *Figure 1G''*) and the two epidermal flanks continue progressing toward the dorsal midline, covering the dorsal opening and eventually fusing completely (*Figure 1—figure supplement 3A''*).

## Inhibition of JNK/Dpp signaling in *Megaselia abdita* arrests dorsal closure but not serosa rupture

Since the JNK and Dpp signaling pathways are known to regulate dorsal closure in *D. melanogaster* embryos, and their impairment results in the failure of dorsal closure and in a dorsal-open phenotype (*Glise and Noselli, 1997*; *Sluss et al., 1996*), we wondered whether JNK and Dpp signaling would also be important regulators of dorsal closure and serosa retraction in *M. abdita* embryos. Using *in situ* hybridization and cuticle preparations, we observed that in wild-type embryos, *M. abdita dpp* (*Mab_dpp*) is expressed along the leading edge of the epidermis (black arrowheads in *Figure 2A*) and progression of dorsal closure results in the deposition of a continuous larval cuticle (*Figure 2A'*), very similar to *D. melanogaster*. Next, we perturbed the *M. abdita* JNK pathway, using gene knockdown by RNA interference (RNAi, see Materials and methods) against *M. abdita bsk* (*Mab_bsk*) in pre-gastrulating embryos. Around 90% of *Mab_bsk* dsRNA-injected embryos developed to at least germband retraction stage (821 out of 922 embryos). *Mab_bsk* RNAi knock-down resulted in both a

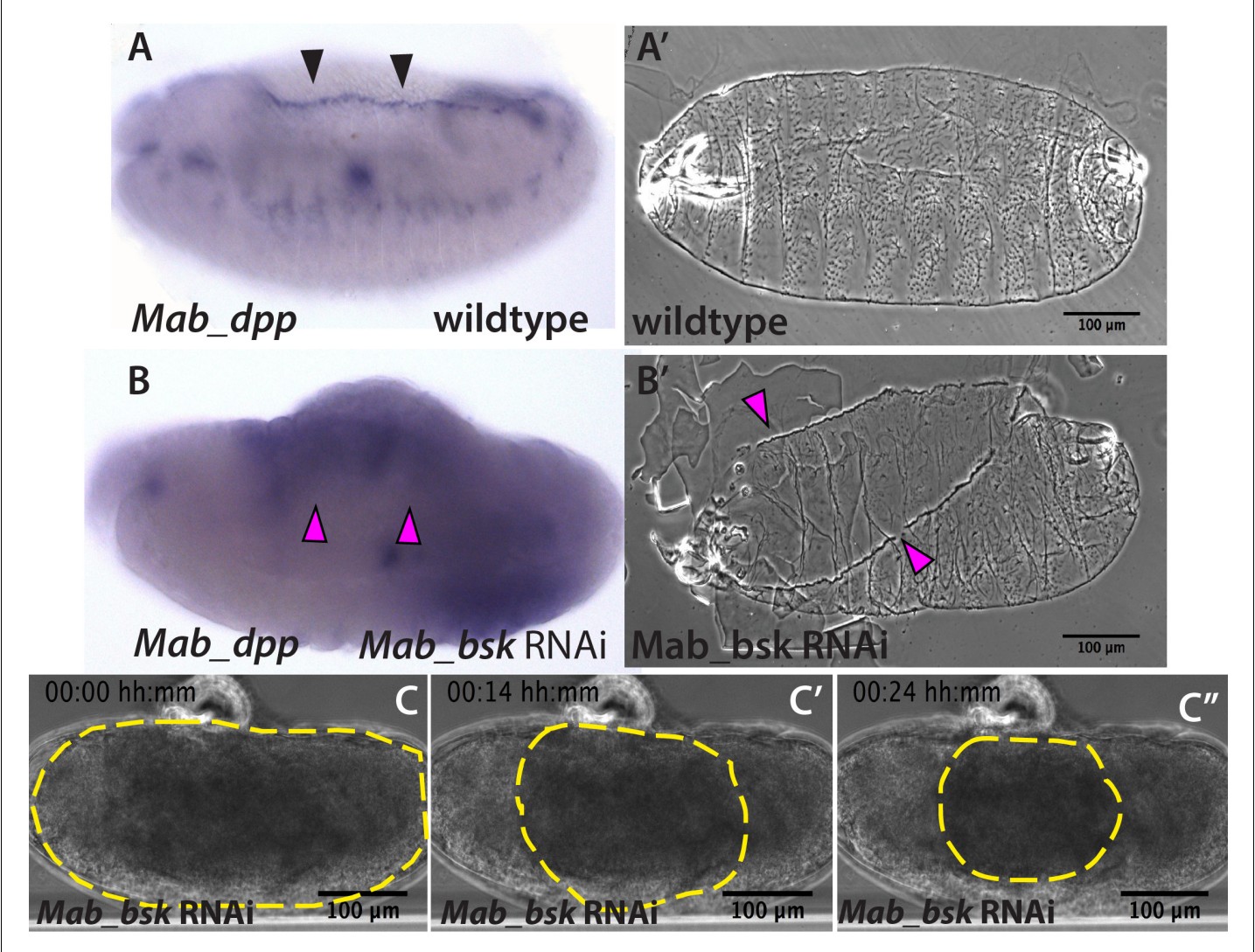

**Figure 2.** JNK (*bsk*) is required for *dpp* expression and the completion of dorsal closure without affecting serosa rupture or retraction in *Megaselia abdita*. (**A**) Wild-type expression of *Mab_dpp* (purple stain, black arrowheads) along the dorsal leading edge of the epidermis in a *M. abdita* embryo (lateral view) during dorsal closure. (**A′**) Cuticle preparation of a wild-type late-stage pre-hatching *M. abdita* embryo (dorsal view). (**B**) RNAi knock-down of *Mab_bsk* abolishes *Mab_dpp* expression at the leading edge of the epidermis (magenta arrowheads) in a *M. abdita* embryo (lateral view). (**B′**) Cuticle preparation of a *Mab_bsk* RNAi late-stage pre-hatching embryo, showing a dorsal-open phenotype (magenta arrowheads). (**C**) Images from a bright-field time-lapse sequence of serosa retraction in a *M. abdita* embryo treated with *Mab_bsk* RNAi (from *Video 3*). Yellow dashed lines show the perimeter of the serosa covering the embryo during retraction. In all embryos and cuticles anterior is to the left and dorsal to the top.

DOI: https://doi.org/10.7554/eLife.33807.010

The following figure supplement is available for figure 2:

**Figure supplement 1.** *Mab_bsk* RNAi results in a lack of dorsal closure progression after serosa rupture.

DOI: https://doi.org/10.7554/eLife.33807.011

disrupted pattern of *Mab_dpp* expression (including a complete absence of expression at the leading edge of the epidermis, magenta arrowheads in *Figure 2B*) and a dorsal-open phenotype in the larval cuticle (magenta arrowheads in *Figure 2B′*) of ~64% of RNAi-injected and developed embryos (528 out of 821 embryos). Both phenotypes are similar to the ones that occur in *D. melanogaster* embryos after JNK/Dpp signaling perturbation. Interestingly, live imaging of RNAi-injected embryos reveals that serosa rupture and retraction still occur (yellow dashed line in *Figure 2C–C′′* and *Video 3*), despite a failure of progression of the epidermal flanks (yellow arrowheads in *Figure 2— figure supplement 1B*; note that the embryo from *Video 3* and *Figure 2C* corresponds to the same

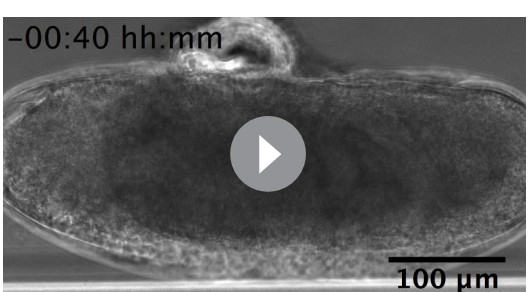

**Video 3.** Knock-down of *Mab_bsk* by RNAi does not prevent serosa rupture and retraction despite preventing epidermal flanks from closing in *Megaselia abdita* embryos. Bright-field time-lapse sequence of serosa retraction in a *M. abdita* embryo injected with *Mab_bsk* dsRNA at early stages of development. Embryo in lateral view. Anterior is to the left, dorsal to the top.

DOI: https://doi.org/10.7554/eLife.33807.012

embryo in *Figure 2—figure supplements 1B*, 24 hr after RNAi injection). In summary, these results indicate that JNK and Dpp signaling regulate progression of dorsal closure in *M. abdita* as in *D. melanogaster* embryos but are not required for the regulation of serosa rupture and retraction.

## Actomyosin contractility in the serosa and an actomyosin epidermal cable are necessary for dorsal closure in *Megaselia abdita* embryos

In *D. melanogaster* embryos, both a JNK/Dpp-dependent contractile epidermal cable and the contraction of the amnioserosa tissue (through actomyosin contractility and volume decrease) power the progression of dorsal closure (*Hutson et al., 2003*; *Kiehart et al., 2000*; *Saias et al., 2015*). The actomyosin cytoskeleton is therefore an essential component in force generation during this process. To gain insights into the structures generating forces during dorsal closure in *M. abdita*, we stained fixed embryos with phalloidin (to reveal F-actin) or a phosphoMyosin antibody. Both stains showed accumulation at the leading edge of the epidermis (green arrowheads in *Figure 3A and B*) and at the surface of serosa cells during internalization (red arrowheads in *Figure 3A and B*). Measurements from time-lapse sequences showed that serosa cell area reduces over time during their accumulation at the dorsal opening, from an average of $212 \pm 52$ μm$^2$ (SD; $n = 20$ cells) to $76 \pm 18$ μm$^2$ (SD; $n = 20$ cells) to $33 \pm$ μm$^2$ (SD; $n = 20$ cells) at 20, 30 and 40 min after serosa rupture, respectively (*Figure 3—figure supplement 1B*). This cell area reduction correlates with an apical accumulation of actin at early stages of serosa internalization (yellow arrowheads in *Figure 1—figure supplement 4a″*, b″ and c″), suggesting an apical constriction mechanism through actomyosin contraction during serosa internalization. In contrast, amnion cells show low levels of actin and myosin, even during late stages of dorsal closure after full serosa internalization (blue arrowhead in *Figure 3A and B*, *Figure 3C″* and *Figure 3—figure supplement 1A*). Closer observation also reveals the presence of actin-enriched filopodia-like extensions protruding from the actomyosin cable (yellow arrowheads in *Figure 3—figure supplement 1C and C′*). RNAi knock-down of *Mab_bsk* strongly reduces the level of actin accumulation at the epidermal leading edge compared to wild-type embryos (white arrows in *Figure 3—figure supplement 2A and B*). This suggests that the embryonic epidermal cable is similar in structure and regulated by the JNK/Dpp signaling pathway in both *M. abdita* and *D. melanogaster* embryos.

A timed sequence of phalloidin-stained *M. abdita* embryos (*Figure 3C–C′″*) reveals further details concerning the dynamics of dorsal closure: upon serosa internalization, the amniotic flanks (devoid of actomyosin) move to the dorsal midline where the two flanks merge (white arrowhead in *Figure 3C″*). Upon merging of amnion flanks, the actomyosin cable propels the epidermal flanks to the dorsal midline where epidermal seaming takes place (yellow arrowheads in *Figure 3C′, C″ and C′″*).

To affect actomyosin-based tissue contractility, we injected *M. abdita* embryos with the Rho kinase (ROCK) inhibitor Y-27632. This drug has been extensively used in *D. melanogaster* to inhibit actomyosin contractility by blocking ROCK activity and, consequently, downstream targets including myosin phosphorylation (*Czerniak et al., 2016*; *Monier et al., 2010*; *Sommi et al., 2011*). Injection of Y-27632 at early stages of *M. abdita* dorsal closure reduces actomyosin accumulation at the epidermal leading edge compared to wild-type embryos (white arrowheads in *Figure 3—figure supplement 2A and C*). The progression of the epidermal leading edge is arrested upon treatment (*Figure 3—figure supplement 2D–D″* and *Video 4*). The internalization of serosa cells is also abolished as observed by the lack of inward bending and apicobasal elongation of this tissue in orthogonal view (*Figure 3—figure supplement 2E–E″* and *Video 5*). These observations indicate that

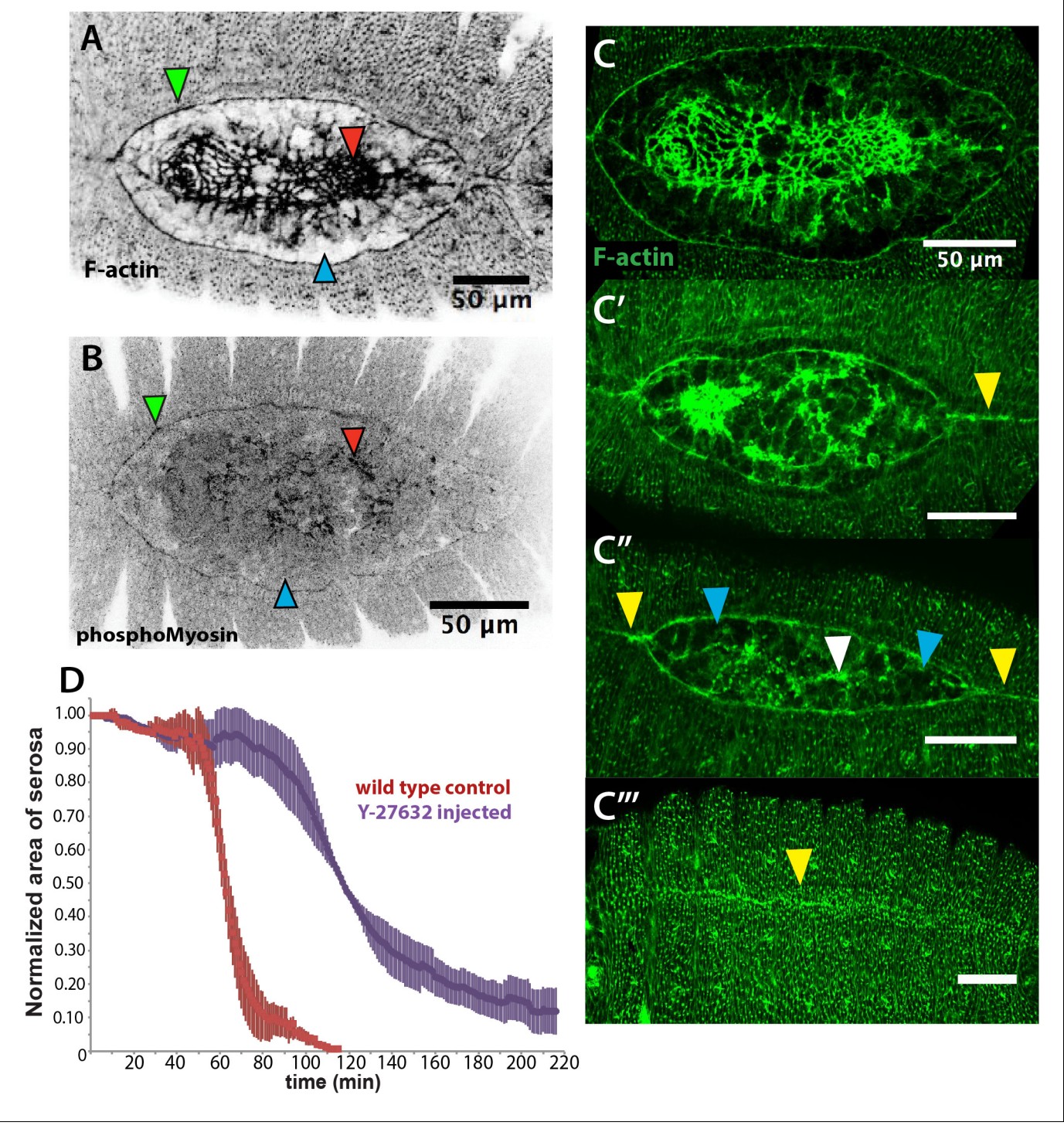

**Figure 3.** An actomyosin-enriched epidermal cable is necessary for the completion of dorsal closure upon contraction, retraction, and internalization of the serosa in *Megaselia abdita*. (**A**) F-actin enrichment (as observed by inverted intensity of phalloidin staining) and (**B**) PhosphoMyosin enrichment (inverted intensity of immunostaining) in the epidermal leading edge (green arrowhead) and internalizing serosa (red arrowhead), but not in the amnion (blue arrowhead) of a *M. abdita* embryo during dorsal closure. (**C–C'''**) Time series of different phalloidin-stained *M. abdita* embryos that show merging of both amniotic and epidermal flanks brought together at the dorsal midline upon serosa internalization. Amniotic merging (blue arrowheads) is visible by a transient accumulation of F-actin (white arrowhead in **C''**). Epidermal merging is mediated by the actomyosin cable (yellow arrowheads). (**D**) Average area of the serosa during retraction in wild-type control (red, *n* = 15 embryos), and contractility-impaired embryos injected with Y-27632 (purple, *n* = 15 embryos). The latter show delayed and incomplete serosal retraction and ingression. Vertical bars represent standard deviations (SD). Embryos from A to D are in dorsal view where anterior is to the left. Embryo in D' is an optical transverse view where dorsal is to the top.

*Figure 3 continued on next page*

*Figure 3 continued*

DOI: https://doi.org/10.7554/eLife.33807.013

The following figure supplements are available for figure 3:

**Figure supplement 1.** Actin accumulates apically in contracting serosa cells and filopodial protrusions are present in the epidermal actin cable of *Megaselia abdita*.

DOI: https://doi.org/10.7554/eLife.33807.014

**Figure supplement 2.** The epidermal actomyosin cable is a contractile structure controlled by JNK (*Mab_bsk*) expression during dorsal closure.

DOI: https://doi.org/10.7554/eLife.33807.015

actomyosin contractility contributes to both the internalization of the serosa and progression of the epidermal leading edge. Interestingly, the kinetics of serosa retraction in Y-27632-injected embryos seemed less affected during the first half of the process than the second half (purple curve in *Figure 3D*), suggesting that actomyosin-based cell contraction is taking place mainly at the final stage of retraction and during serosa internalization.

Taken together, these results indicate that actomyosin-based contractility within the serosa and the cable surrounding the dorsal opening are required for the progression of dorsal closure in *M. abdita* embryos.

## A microtubule-based seaming of the extraembryonic amnion is required for dorsal closure in *Megaselia abdita* embryos

In addition to an actomyosin cable, the microtubule cytoskeleton is known to be essential during the last step of dorsal closure in *D. melanogaster*. In epidermal cells at the leading edge, microtubules align in apical bundles parallel to the dorsoventral axis and protrude into stable filopodia. These aligned epidermal microtubules are required for proper epidermal seaming and their depolymerization leads to defects in seaming and incomplete closure in *D. melanogaster* (*Jankovics and Brunner, 2006*). We investigated whether such a microtubule configuration is observable during dorsal closure in *M. abdita* as well. Staining fixed *M. abdita* embryos at different stages of *late* dorsal closure with a β-tubulin antibody, we observed that microtubules in the epidermis also orient toward the dorsal midline (*Figure 4—figure supplement 1A and B*) and protrude from the epidermal leading edge (white arrowheads in *Figure 4—figure supplement 1B*) forming apical bundles (*Figure 4—figure supplement 1B′*). This epidermal microtubule alignment is maintained throughout epidermal flank advancement and during epidermal seaming (*Figure 4—figure supplement 1C–C′′′*). Interestingly, we find a similar alignment of microtubules in the extraembryonic amnion, where microtubule bundles localize apically (*Figure 4B′*) and orient toward the internalizing serosa (blue arrowheads in *Figure 4A and B* and *Figure 4—figure supplement 1A*). This apical microtubule alignment in the amnion seems to follow cell elongation. To support this observation, we estimated apical cell surface area by performing a Voronoi tessellation around the nuclei of extraembryonic cells. We could observe that amniotic cells present an elongated apical cell surface area ($285 \pm 81\ \mu m^2$, SD; $n = 12$ cells; blue arrowheads in *Figure 4—figure supplement 1D*) compared to serosa cells ($57 \pm 23\ \mu m^2$, SD; $n = 108$ cells; red arrowheads in *Figure 4—figure supplement 1D*). Both apical microtubule alignment and elongated apical cell surface area in the amnion are maintained during serosa internalization and merging of amnion cells from opposite flanks (blue arrowheads in *Figure 4B* and *Figure 4—figure supplement 1C–C′*). Amnion

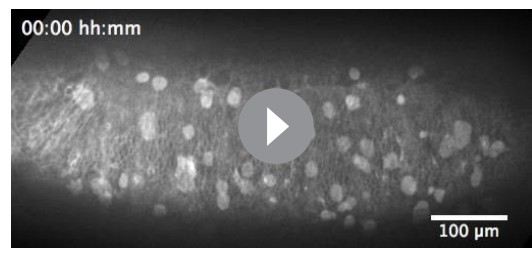

**Video 4.** Injection of the Rho kinase inhibitor Y-27632 prevents dorsal closure and slows down serosa retraction in *Megaselia abdita*. Time-lapse sequence of dorsal closure in a FM 4–64-labeled *M. abdita* embryo injected with the Rho kinase (ROCK) inhibitor Y-27632, which downregulates actomyosin-based contractility. Note the slow rupture and retraction of the serosa, and the failure of the epidermal flanks to advance and close. Embryo in lateral view where anterior is to the left, dorsal to the top. Note that the static stains observed are stains on the vitelline envelope arising from embryo treatment prior to imaging.

DOI: https://doi.org/10.7554/eLife.33807.016

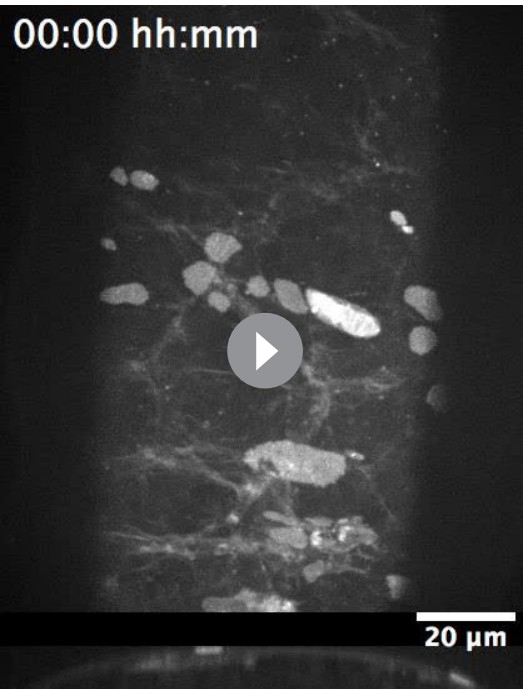

**Video 5.** Injection of the Rho kinase inhibitor Y-27632 prevents internalization of the serosa in *Megaselia abdita* embryos. Time-lapse sequence of a dorsal (top) and orthogonal view (bottom) of serosa rupture in an FM 4–64-labeled *M. abdita* embryo. Injection of the Rho kinase (ROCK) inhibitor Y-27632 (to downregulate actomyosin-based contractility) prevents internalization of the serosa. The inward bending and apicobasal cell elongation of the extraembryonic tissue into the yolk is not observed in orthogonal view. In the dorsal view, anterior is to the top. In the orthogonal view, dorsal is to the top. Note that the static stains observed are stains on the vitelline envelope arising from embryo treatment prior to imaging.
DOI: https://doi.org/10.7554/eLife.33807.017

microtubule alignment is subsequently lost after the opposite amnion flanks meet at the dorsal midline (blue arrowhead in *Figure 4—figure supplement 1C''*).

Since microtubule alignment is necessary for epidermal seaming during dorsal closure in *D. melanogaster*, we reasoned that a similar microtubule alignment in the amnion of *M. abdita* embryos could indicate epithelial amniotic fusion through seaming in this species. To follow the last steps of *M. abdita* dorsal closure more closely, we imaged FM 4–64-labeled embryos during the *late* stage of dorsal closure (*Video 6*). We observed that upon serosa internalization, the opposing amniotic flanks of *M. abdita* embryos meet and fuse at the dorsal midline, suggesting an amniotic seaming process (blue-shaded area in *Figure 4C,C' and C''*). Immediately after amniotic seaming occurred, the epidermal flanks progressed dorsally and also seamed on top of the continuous amnion (green dashed line in *Figure 4C,C' and C''*). An estimation of seaming velocities in the amnion and epidermis ($8.1 \pm 2$ µm/min and $3.8 \pm 1.7$ µm/min, SD, $n = 5$ embryos, respectively) shows a variation in speed between the two processes that could result from a difference in the mechanical properties of the fusing epithelia or the seaming angle at each canthi.

To investigate the functional role of microtubules in amnion cell elongation and seaming, we injected embryos with the microtubule depolymerizing drug colcemid (*Jankovics and Brunner, 2006*). Injection of colcemid does not perturb the kinetics of serosa retraction (*Figure 4—figure supplement 2A*), actin accumulation in serosa cells (red arrowheads in *Figure 4—figure supplement 2B and B'*), straightening of the epidermal leading edge (white arrowheads in *Figure 4D*), or actin cable formation (white arrowheads in *Figure 4—figure supplement 2B and B'*). An orthogonal view from a time-lapse sequence also shows that the initial inward bending and apicobasal elongation of the serosa cells during internalization occurs in both wild type and colcemid injected embryos (*Video 7* and magenta dashed lines in *Figure 1—figure supplement 3C–C'* and *Figure 4—figure supplement 2C–C'*).

In contrast, dorsal closure arrests during the internalization of the serosa (red arrowhead in *Figure 4D''*) and does not progress toward epithelial seaming (*Video 8* and red curve in *Figure 4E*). We observe that microtubule polymerization does not occur in colcemid-injected embryos (blue and white arrowheads for amnion and epidermal cells, respectively, in *Figure 4—figure supplement 2D* compared to 2D') and that amnion cells initially elongate toward the dorsal midline as in wild-type conditions, although they relax and retract from the amnion merging site (*Figure 4—figure supplement 4A* and *Video 9*). Thus, microtubule depolymerization does not affect amnion cell elongation and dorsal convergence but rather a subsequent amniotic seaming. In order to test the role of the microtubule assembly, specifically in the amnion, we deactivated the depolymerizing effect of colcemid treatment on dorsal closure progression with UV light. UV-irradiation was performed in a region between the epidermis and the internalizing serosa, corresponding to the amnion (magenta area in *Figure 4—figure supplement 3A''* and *Video 10*). In UV-irradiated embryos, dorsal closure

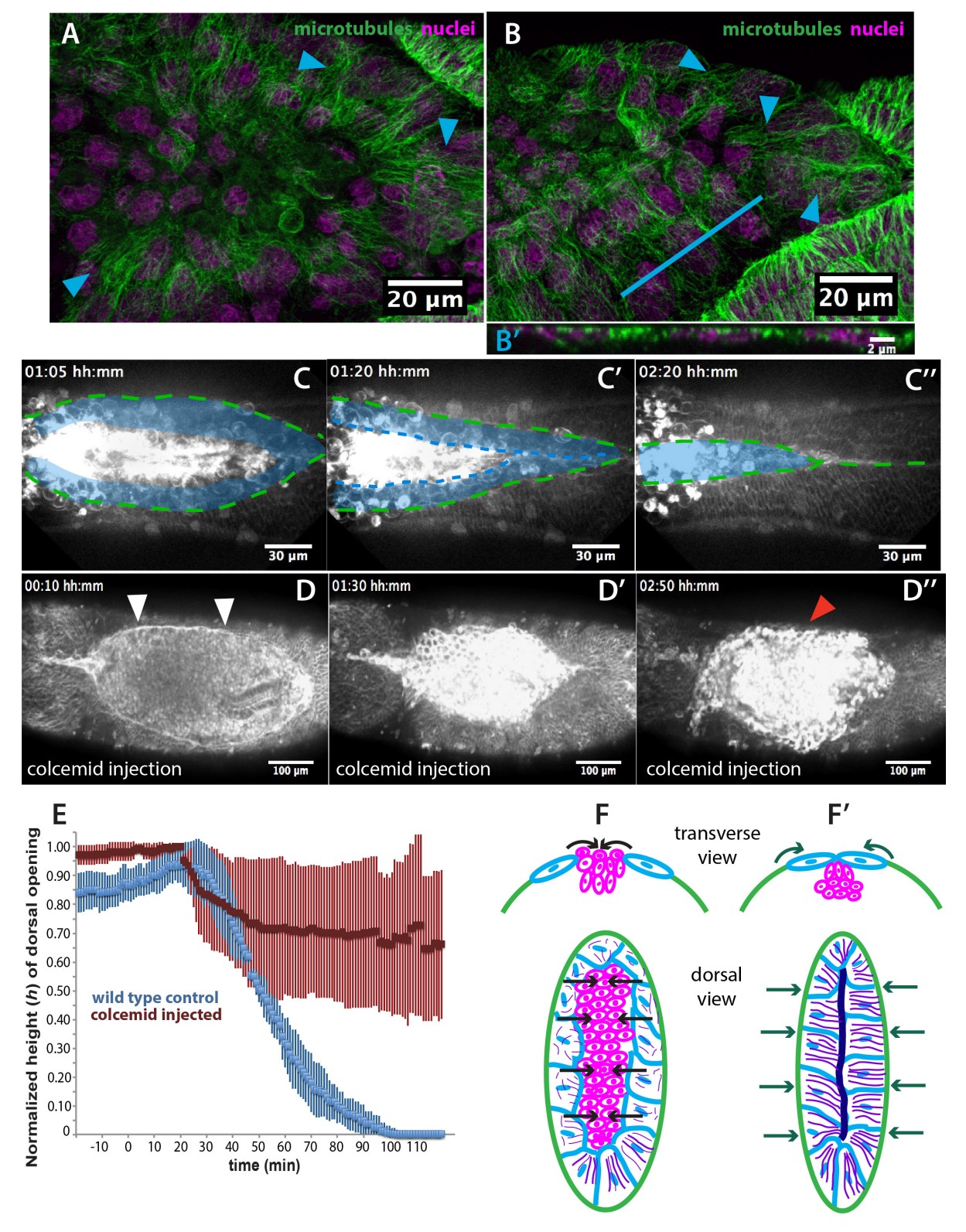

**Figure 4.** Microtubule-dependent seaming of the extraembryonic amnion is required for subsequent epidermal seaming during dorsal closure in *Megaselia abdita*. (A) Confocal projections of microtubules in the extraembryonic amnion orienting toward the internalizing serosa during early amniotic seaming. (B) As amniotic seaming progresses, microtubules maintain alignment, and become localized apically, as revealed by a transverse view (B′), obtained by orthogonal stack re-slicing along the blue line in B. Staining against β-tubulin is shown in green, DAPI nuclear counterstain in

*Figure 4 continued on next page*

*Figure 4 continued*

magenta. (C) Time-lapse sequence of amniotic seaming (blue dashed line and blue shaded area) followed by epidermal seaming (green dashed line) along the dorsal midline in a *M. abdita* embryo injected with the fluorescent lipophilic dye FM 4–64 (from ***Video 6***). (D) Time-lapse sequence of dorsal closure in a colcemid-treated embryo to induce microtubule depolymerization. After serosa retraction the process is arrested and closure fails. The embryo is labeled by FM 4–64 (from ***Video 8***). White arrowheads in D show the proper straightening of the epidermal leading edge in colcemid-injected embryos. Red arrowhead in D'' indicates impaired serosa internalization. (E) Relative changes in height (*h*) of dorsal opening during in wild-type control (blue, *n* = 15 embryos) and embryos injected with colcemid (red, *n* = 15 embryos) to depolymerize microtubules. Vertical bars show standard deviations (SD). These measurements reveal failed epidermal leading edge progression in colcemid-treated *M. abdita* embryos. (F) Schematics depicting the transverse view (top) and dorsal view (bottom) of embryos during serosa cell internalization, initiation of amnion cell elongation, and microtubule alignment (purple) toward the dorsal midline. Black arrows indicate the direction of amnion progression. (F') Amnion cells show alignment of apical microtubule bundles toward the dorsal midline, where the two amniotic flanks meet, and amniotic seaming occurs (dark blue line). This is followed by the progression of the epidermal leading edge (dark green arrows), which results in epidermal seaming and completion of dorsal closure (not shown). Serosa in magenta, amnion in blue and embryonic epidermis in green. All embryo images show dorsal views where anterior is to the left.

DOI: https://doi.org/10.7554/eLife.33807.018

The following figure supplements are available for figure 4:

**Figure supplement 1.** Microtubule alignment is present in both the epidermis and the amnion and correlates with enlarged amniotic cells in *Megaselia abdita*.

DOI: https://doi.org/10.7554/eLife.33807.019

**Figure supplement 2.** Colcemid treatment prevents microtubule polymerization without affecting serosa retraction or epidermal actomyosin cable in *Megaselia abdita*.

DOI: https://doi.org/10.7554/eLife.33807.020

**Figure supplement 3.** UV-deactivation of colcemid in the amnion region allows dorsal closure progression in treated *Megaselia abdita* embryos.

DOI: https://doi.org/10.7554/eLife.33807.021

**Figure supplement 4.** Landmarks of amniotic seaming during dorsal closure progression in *Megaselia abdita*.

DOI: https://doi.org/10.7554/eLife.33807.022

**Figure supplement 5.** Schematics of dorsal closure in *Megaselia abdita*.

DOI: https://doi.org/10.7554/eLife.33807.023

progressed further, reducing the height (*h*) of the dorsal opening, compared to colcemid-treated embryos, although it did not progress to the extent of wild-type control embryos (***Figure 4—figure supplement 3B***). Taken together, these observations indicate that the microtubule cytoskeleton in the amniotic tissue is required for seaming of the amnion and the completion of dorsal closure.

In summary, the results presented in this section indicate the presence of subsequent microtubule-based seaming processes in both the amnion and the epidermis during dorsal closure in *M. abdita*. First, opposing amniotic flanks fuse at the dorsal midline upon serosa ingression, followed by epidermal seaming (see schematics in ***Figure 4F and F'*** and lower panels of ***Figure 4—figure supplement 4b–b''***). Both processes are necessary for the completion of epidermal seaming and dorsal closure, to result in a continuous embryonic epidermal sheet. Our results indicate that serosa contraction contributes to bringing amniotic flanks into close proximity and that amniotic seaming is a microtubule-dependent event.

## Discussion

In this study, we provide a detailed characterization of epithelial fusion during dorsal closure in the non-drosophilid scuttle fly *M. abdita*. In this species, dorsal closure involves three different tissues: the embryonic epidermis, as well as the extraembryonic amnion and serosa. Dorsal closure in *M. abdita* occurs in three distinct phases: (i) rupture and retraction of the extraembryonic serosa surrounding the embryo, (ii) concurrent contraction of both an epidermal actomyosin cable and the serosal tissue in the dorsal region of the embryo leading to internalization of the serosa into the dorsal opening, and (iii) successive seaming processes fusing first the amnion and then the epidermis. Even though genetic regulation of dorsal closure by the JNK and Dpp signaling pathways appears to be conserved between *M. abdita* and *D. melanogaster*, the sequence of morphogenetic rearrangements is very different between the two species. These differences, however, result in the same output: a continuous epidermal layer covering the dorsal region of the embryo.

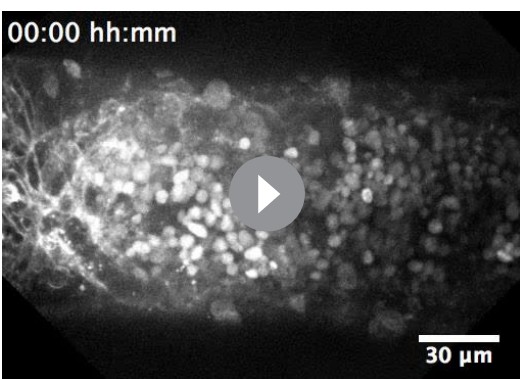

**Video 6.** Amniotic seaming followed by epidermal seaming during the late stage of dorsal closure in *Megaselia abdita*. Time-lapse sequence of the final stage of dorsal closure in a *M. abdita* embryo fluorescently labeled with FM 4–64. The amnion flanks are brought together and seamed at the dorsal midline upon serosa ingression. This process is followed by seaming of the epidermal flanks at the dorsal midline. Both seaming processes initiate at the posterior end of the embryo. Embryo in dorsal view. Anterior is to the left. Note that the static stains observed are stains on the vitelline envelope arising from embryo treatment prior to imaging.
DOI: https://doi.org/10.7554/eLife.33807.024

In *M. abdita*, the serosa encloses the whole embryo. It is apposed to the amnion, which in turn is apposed to the epidermis at the edge of the dorsal opening (schematics in *Figure 1B and B'*) (see also *Rafiqi et al., 2008*). The rupture of the serosa is the first step of a series of complex morphogenetic events (see *Figure 4—figure supplement 5*). Although the initiation signal for serosal rupture is not yet known, we can discard a purely mechanical trigger since injection of the embryo prior to dorsal closure did not induce serosal rupture and global retraction, despite resulting in a small wound and a slight retraction of the tissue around the injection site. In addition, rupture still occurs in embryos injected with Rho-kinase (ROCK) inhibitor, which reduces actomyosin contractility. Lastly, rupture always initiates at a very specific ventral-posterior site. Taken together, these observations indicate that rupture is not triggered exclusively by global straining and non-autonomous forces applied to the serosa tissue. Instead, rupture seems to be triggered by a specific localized cue.

Upon rupture, the remaining serosal tissue retracts and constricts dorsally through an actomyosin-dependent mechanism, in a way similar to serosa rupture and retraction in the beetle *T. castaneum* (*Hilbrant et al., 2016*; *Panfilio et al., 2013*). The retracting serosa then internalizes into the dorsal opening of *M. abdita*. Concomitant with serosal internalization, a JNK/Dpp-dependent actomyosin cable forms at the epidermal leading edge of *M. abdita* embryos. It promotes the advancement of the opposing epidermal flanks toward the dorsal midline, and the eventual seaming of the two flanks. This stage of dorsal closure occurs in a similar fashion to *D. melanogaster*, but differs in comparison with *T. castaneum*, where no actomyosin epidermal cable appears to be involved in epidermal flank advancement (*Panfilio et al., 2013*).

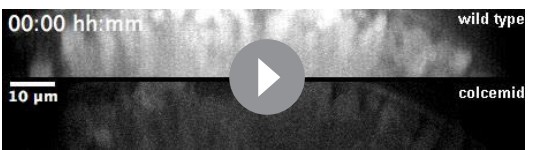

**Video 7.** The initial stages of serosa internalization in colcemid-injected embryos of *Megaselia abdita* occur as in wild-type embryos. Time-lapse sequence in orthogonal view of dorsal closure during serosa internalization in FM 4–64-labeled *M. abdita* embryos. The inward bending of the extraembryonic tissue and apicobasal cell elongation into the yolk is observed in wild-type control (top) and colcemid-injected embryos (bottom). Injection of colcemid induces microtubule depolymerization and impairs the late stages of dorsal closure (see *Video 8*). Dorsal is to the top.
DOI: https://doi.org/10.7554/eLife.33807.025

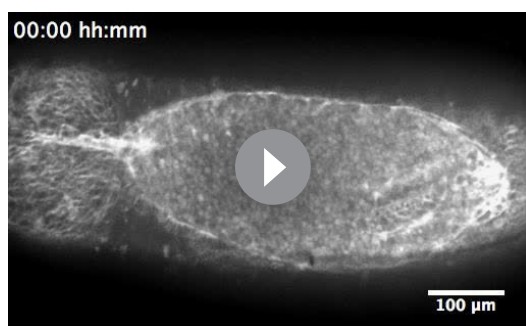

**Video 8.** Colcemid injection prevents dorsal closure in *Megaselia abdita*. Time-lapse sequence of impaired dorsal closure in an FM 4–64-labeled *M. abdita* embryo after injection of colcemid to induce microtubule depolymerization. The initial stages of dorsal closure (straightening of the epidermal leading edge and serosa rupture and retraction) are not affected. The process of dorsal closure is aborted during the late stages (amniotic seaming and epidermal seaming). Embryo in dorsal view. Anterior is to the left.
DOI: https://doi.org/10.7554/eLife.33807.026

In contrast to *D. melanogaster*, dorsal closure in *M. abdita* involves an additional amniotic seaming process. Our experimental data indicate that amniotic seaming is microtubule-dependent and essential for dorsal closure to occur. Thus, similar to epidermal seaming in *D. melanogaster* embryos, where microtubules align dorsoventrally prior to tissue fusion (*Jankovics and Brunner, 2006*), the two sequential amniotic and epidermal seaming processes in *M. abdita* also involve a dorsoventral alignment of microtubules.

Why microtubules align in this way remains unclear. One possible scenario is that shape elongation of amniotic cells toward the retracting and internalizing serosa could promote microtubule reorientation in the direction of contractile cells. Interestingly, cellular fusion in the developing trachea of *D. melanogaster* involves cell elongation and microtubules orientation toward the site of fusion (*Kato et al., 2016*). Elongation of cells toward a contractile tissue also occurs during gastrulation in *D. melanogaster* (*Rauzi et al., 2015*) and neural tube closure in the chordate *Ciona intestinalis* (*Hashimoto et al., 2015*). It is not known whether microtubule alignment also occurs in the latter processes to promote epithelial fusion. If this is the case, the microtubule-dependent seaming that we describe might reflect a common mechanism for epithelial tissues to fuse.

It remains unclear whether microtubule-dependent epithelial seaming is a process that can generate forces contributing to dorsal closure. In the case of *D. melanogaster*, laser-ablation of epidermal canthi (*i.e.* the epidermal corners where opposing epidermal flanks meet) slows down the last stages of dorsal closure (*Wells et al., 2014*). However, F-actin-enriched epidermal seaming still occurs between the opposing leading edges of the epidermis despite the removal of the canthi.

Dorsal closure in embryos of *M. abdita* presents two sequential seaming events that share a common feature: transient microtubule reorganization. It would be interesting to explore if the cytoskeletal basis (*i.e.* microtubule reorganization) of epithelial seaming events is conserved in other insect species with a three-tissue system of dorsal closure, for example *T. castaneum*.

Our work suggests that the evolutionary transition from a three-tissue to a two-tissue system of dorsal closure not only involves the reduction of extraembryonic tissue, for example from distinct amnion and serosa to a fused amnioserosa (see *Horn et al., 2015*; *Rafiqi et al., 2008*; *Rafiqi et al., 2010*; *Schmidt-Ott and Kwan, 2016*). In addition, it requires changes in epidermal progression and seaming events. Further development of imaging and molecular tools in *M. abdita* will help us better understand the subcellular, cellular, and tissue dynamics that led to this evolutionary transition.

In the case of dipteran dorsal closure, it appears that the evolutionary modulation of tissue remodeling is mainly driven by morphological rearrangements rather than large changes in gene expression. This study provides the first detailed analysis of tissue anatomy and dynamics complemented with gene expression assays to understand the evolution of a morphogenetic process. In

**Video 9.** Colcemid injection impairs amniotic flank seaming during dosal closure in *Megaselia abdita*. Time-lapse sequence in lateral view of the latest stages of dorsal closure in FM 4–64-labeled *M. abdita* embryos. Amnion cell elongation can be observed in a wild-type embryo (top) resulting in the seaming of the amniotic flanks followed by seaming of the epidermal flanks. In colcemid-injected embryos (bottom), amnion cells initially elongate but fail to maintain elongation and retract from the amniotic merging at the dorsal midline. Note that the static stains observed are stains on the vitelline envelope arising from embryo treatment prior to imaging.
DOI: https://doi.org/10.7554/eLife.33807.027

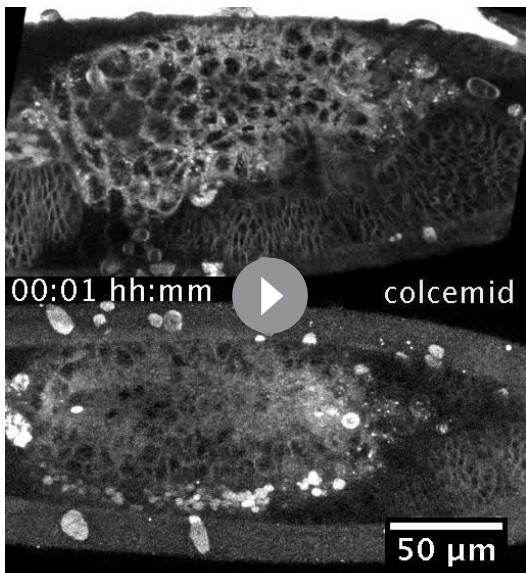

**Video 10.** UV-deactivation of colcemid in the amnion region of treated *Megaselia abdita* embryos rescues amnion seaming. Time-lapse sequence in dorsal view of dorsal closure in FM 4–64-labeled *M. abdita* embryos during serosa accumulation and internalization. Colcemid-injected embryos do not complete dorsal closure (top). UV-deactivation of colcemid in a region of interest (ROI) between the epidermal flanks and the internalizing serosa (bottom) rescues serosa internalization.

DOI: https://doi.org/10.7554/eLife.33807.028

this respect, dipterans provide a powerful model to understand the interplay between tissue rearrangement and gene expression during the evolution of development.

## Materials and methods

### Fly husbandry, embryo collection, cloning procedures, and RNAi knock-down

Our *M. abdita* fly culture was maintained as previously described (*Rafiqi et al., 2011a*). Embryos were collected at 25°C for 4 hr, and then incubated at 19°C until they reached stages 13–15, corresponding to dorsal closure as described in *Wotton et al. (2014)*. *Mab_bsk* was cloned using sequence data from a published early embryonic transcriptome (http://diptex.crg.es; gene ID: *Mab_bsk*: MK10) (*Jiménez-Guri et al., 2013*). Briefly, open-reading frames (ORFs) were PCR-amplified based on cDNA from 0 to 5 hr-old *M. abdita* embryos. Amplified fragments were cloned into PCRII-TOPO (Invitrogen, Carlsbad, CA) or pGEM-T (Promega, Madison, WI) vectors using the following specific primers (5'/3'): *Mab_bsk*, TGCCCG TCATCAGTTTTACA and GACGACGCGGGAC TACTTTA. dsRNA was performed using the Ambion MEGAscript kit (Life Technologies, Carlsbad, CA). The following specific primers (5'/3') containing a T7 promoter sequence at their 5' end were used: *Mab_bsk*, GGTGGGCGACACAAGATT and AAACAGGCATCGGGGAAT. RNAi injection was performed using previously published protocols (*Rafiqi et al., 2008*; *Rafiqi et al., 2010*; *Rafiqi et al., 2011d*; *Wotton et al., 2015*). Dechorionated embryos were injected prior to gastrulation at a concentration of 5 μM for *Mab_bsk*, then incubated at 25°C. The injected dsRNA construct comprised 798 nucleotides (base pairs 369–1166 of the ORF) for *Mab_bsk*.

### *In situ* hybridization, immunohistochemistry and cuticle preparations

In situ hybridization in heat-fixed *M. abdita* embryos was performed according to a previously published protocol from *D. melanogaster* (*Crombach et al., 2012*). Digoxigenin-labeled *Mab_dpp* probe is from *Jiménez-Guri et al. (2013)*. Fixation, devitellinization and immunostaining of *M. abdita* embryos were performed as previously described (*Rafiqi et al., 2011c*; *Rafiqi et al., 2012*) with slight modifications. Briefly, embryos undergoing dorsal closure were dechorionated and fixed for 25 min in heptane and PEMS (100 mM PIPES, 2 mM EGTA and 1 mM $MgSO_4$, pH 6.9), in a 3:1 PEMS:methanol solution, and a final concentration of 6.5% formaldehyde. Embryos were postfixed and hand devitellinized as described (*Rafiqi et al., 2012*). Microtubules were stained using a monoclonal primary antibody (mouse) against β-tubulin (E7, Developmental Studies Hybridoma Bank) at a dilution 1:100, and a secondary antibody conjugated to Alexa 488 dye (Invitrogen, Carlsbad, CA) at a dilution of 1:1000. For phalloidin staining, embryos were fixed for 1 hr using PEMS and a final concentration of 8% formaldehyde, hand-devitellinized as described for *D. melanogaster* embryos (*Fernández et al., 2007*; *Kaltschmidt et al., 2002*; *Rothwell and Sullivan, 2000*), and incubated with phallodin-Alexa488 or phalloidin-Alexa563 (Invitrogen, Carlsbad, CA) at a dilution of 1:200 for 1 hr. When double-staining against phalloidin and microtubules, embryos were fixed, hand-devitellinized, and stained for phalloidin first, followed by incubation with the β-tubulin primary and

secondary antibodies as above. Nuclei were counterstained using DAPI (1:1000). Embryos were washed in PBT (PBS, with 0.1% Triton X-100), and mounted using ProLong Gold Antifade (Invitrogen, Carlsbad, CA). Cuticle preparations of *M. abdita* embryos were performed as previously described (*Rafiqi et al., 2011b*) with slight modifications. Briefly, embryos were fixed and hand-devitellinized before preparing and mounting the cuticles. Images of cuticle preparations were taken using a phase-contrast microscope.

## Microscopy, live ilmaging, pharmacology, UV irradiation and image processing

Time-lapse imaging was performed with dechorionated *M. abdita* embryos. RNAi-injected embryos were imaged using a Zeiss Cell Observer with a controlled temperature chamber at 25°C and phase contrast settings. For fluorescence imaging, wild type embryos at dorsal closure stage were desiccated for 5 min, aligned, oriented, and immobilized on a coverslip with heptane glue, covered with halocarbon oil and injected at the embryo poles with 1 mM (needle concentration) of the lipophilic dye FM 4–64 (Molecular Probes). Embryos were imaged at room temperature using an Andor Revolution XD spinning-disk confocal microscope. ROCK inhibitor Y-27632 (Sigma-Aldrich, St. Louis, MO) and colcemid (Santa Cruz Biotechnology, Santa Cruz, CA) were prepared to 10 mM and 500 µg/ml (needle concentration), and also injected at the embryo poles during dorsal closure stage. Control embryos were injected with water or DMSO, respectively. Final needle concentrations of dye and/or drugs were prepared in injection buffer (10 mM HEPES, 180 mM NaCl, 5 mM KCl and 1 mM $MgCl_2$, pH 7.2), and delivered to the interstitial space formed between the serosa and the embryo. Confocal projections of ~10 *z*-stack images (1 µm spacing) were used to generate time-lapse sequences in dorsal view. Orthogonal views from time-lapse live imaging were obtained by reslicing confocal *z*-stacks of 0.25 µm spacing. The height (*h*) of the dorsal opening is the maximum perpendicular distance from the dorsal midline to the epidermal leading edge (*Hutson et al., 2003*). The changes in area of the serosa covering the embryo over time were determined by approximating the embryonic shape using an ellipse and resizing manually to follow the serosa edge on one lateral side during retraction, assuming that serosa retraction occurs symmetrically on both sides of the embryo after rupture along the ventral midline. Detection of serosal edge morphology from bright-field time-lapse sequences was performed by subtracting images at time *t* + 1 from images at time *t*. This operation rendered the contour of the retracting serosa visible. The identification of extraembryonic tissues was performed using nuclear anatomy, staining profiles and *z* position of the nuclei in confocal stack images obtained from fixed embryos labeled with DAPI. Staining profiles, nuclear areas and *z* position in the embryo were measured in 150 cells for each cell type (serosa and amnion) from 15 different embryos. UV irradiation experiments to deactivate colcemid were performed as follows: dechorionated, desiccated, FM 4–64-labeled and colcemid-injected *M. abdita* embryos were immobilized in heptane glue, mounted in halocarbon oil, and imaged dorsally in an inverted Leica TCS SP5 laser-scanning confocal microscope in resonant scanner mode. Colcemid injection was performed during dorsal ridge fusion and prior to serosa rupture. Imaging acquisition started ~30 min after injection. A region of interest (ROI) was selected comprising an area between the epidermal flanks and the internalizing serosa, corresponding to the extraembryonic amnion. The ROI was scanned for at least 30 s using a 405 nm UV laser and imaging was resumed after irradiation. Fixed, immunostained embryos were imaged as follows: images were acquired using an inverted Leica TCS SP5 laser-scanning confocal microscope. All post-acquisition image processing and analysis was done using ImageJ software (NIH). For Voronoi analysis, the center of mass of cell nuclei was detected manually with Fiji and used as seed for Voronoi tessellation with Matlab.

## Acknowledgements

We would like to thank Eva Jiménez-Guri, Karl Wotton, and Arturo D'Angelo for reagents and invaluable technical advice during the development of the project. We thank Steffen Lemke for providing training and technical advice as well as Jordi Casanova and Petra Stockinger for discussions and critical reading of the manuscript. All confocal imaging was done at the CRG Advanced Light Microscopy Unit. JJF-Z was supported by a CRG International Interdisciplinary Postdoctoral Programme (INTERPOD) fellowship, co-funded by Marie Curie Actions. We acknowledge support from the Spanish Ministry of Economy and Competitiveness to the EMBL partnership, 'Centro de Excelencia

Severo Ochoa', the, Plan Nacional BFU2015-68754-P (MINECO) and the CERCA programme/Generalitat de Catalunya.

## Additional information

### Funding

| Funder | Grant reference number | Author |
| --- | --- | --- |
| Ministerio de Economía y Competitividad | BFU2015-68754-P | Jérôme Solon |

The funders had no role in study design, data collection and interpretation, or the decision to submit the work for publication.

### Author contributions

Juan Jose Fraire-Zamora, Conceptualization, Formal analysis, Investigation, Methodology, Writing—original draft, Writing—review and editing; Johannes Jaeger, Conceptualization, Supervision, Resources, Funding acquisition, Writing—review and editing; Jérôme Solon, Conceptualization, Formal analysis, Supervision, Resources, Funding acquisition, Writing—review and editing

### Author ORCIDs

Juan Jose Fraire-Zamora ⓘD http://orcid.org/0000-0003-2870-0140
Jérôme Solon ⓘD https://orcid.org/0000-0001-9967-9794

### Decision letter and Author response

Decision letter https://doi.org/10.7554/eLife.33807.036
Author response https://doi.org/10.7554/eLife.33807.037

## Additional files

### Supplementary files

• Transparent reporting form
DOI: https://doi.org/10.7554/eLife.33807.029

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
