## [Decision Letter]

Thank you for submitting your article "Two consecutive microtubule-based epithelial seaming events mediate dorsal closure in the scuttle fly *Megaselia abdita*" for consideration by *eLife*. Your article has been evaluated by K VijayRaghavan (Senior Editor) and three reviewers, one of whom is a member of our Board of Reviewing Editors. The reviewers have opted to remain anonymous.

The reviewers have discussed the reviews with one another and the Reviewing Editor has drafted this decision to help you prepare a revised submission.

Three reviews have now been obtained for the manuscript "Two consecutive microtubule-based epithelial seaming events mediate dorsal closure in the scuttle fly *Megaselia abdita*". As you will see from their individual comments, all three reviewers agree that the manuscript addresses an interesting question. However, all have substantial concerns about the conclusions of the manuscript and suggest ways in which the conclusions must be strengthened. Their suggestions are summarized below.

i) The authors must provide better data to support their claims on tissue topology (serosal ingression, serosa amnion connection), novel cell behaviours (two seaming/fusion events, cell behaviours accompanying ingression) cytoskeletal organisation (actin enrichment at two edges and microtubule reorganisation in the amnion and epidermis) and gene expression patterns. Cross sectional views that support the schematic drawings provided in Figure 1 will be valuable. The temporal evolution of cell biological and cytoskeletal changes must also be better documented even if this comes from fixed rather than live embryos.

ii) The mechanistic claims made are too strong and not backed by the present set of experiments which lack tissue specificity. They must be toned down or backed up by more local perturbations as for example by caged compounds or laser ablation.

iii) The Discussion must be improved to better illustrate the importance of the work and must make references to relevant work in *Drosophila, Tribolium* and *Megaselia*.

*Reviewer #1:*

In this manuscript, Solon and colleagues examine the conservation of signaling and cytoskeletal mechanisms accompanying dorsal closure, a model for epithelial fusion, in the scuttlefly, a cycloraphan, non-Drosophilid dipteran insect, in which the process requires three tissues (including the epidermis and separate amnion and serosa) rather than two (including the epidermis and amnioserosa). The issue being addressed is whether the separation of the amnion and serosa, a common feature of many insects, results in closure that is mechanistically or molecularly different from closure in insects like Drososphila in which they are combined into one. Using a combination of gene expression analysis, cell biology and live imaging, the authors i) document differences (with *Drosophila*) in the tissue dynamics accompanying dorsal closure (which they show includes serosal rupture, retraction and ingression), ii) identify cytoskeletal changes that accompany it (including two sites/edges of actin enrichment and two sites of microtubule orientation rather than one), iii) suggest the involvement of actomyosin based contractility in the serosa and epidermis and microtubule orientation in the formation of the epidermal and amniotic seams and iv) provide evidence for the conservation of *JNK* signalling in the epidermis. They also suggest that both the amnion and the epidermis contribute to seam formation and closure.

Dorsal closure in *Drosophila* has served as a good model to address the conservation of mechanisms underlying epithelial fusion across tissues and organisms. It has also been the focus of studies in evo-devo, particularly with respect to the origin of insect extraembryonic membranes and their indispensable contribution to embryonic patterning. How the two separated extraembryonic membranes interact physically and functionally during dorsal closure has also been the subject of an interesting recent paper on dorsal closure in the flour beetle, *Tribolium* published in *eLife* (Hilbrant et al., *eLife*2016). In this context, I think the manuscript presents some new and interesting findings in an emerging model organism, in particular, the quantitative analysis of tissue dynamics, a suggestion of their force contributions, and involvement of the microtubule cytoskeleton in the formation of two seams rather than one. While the manuscript uses genetic perturbations and chemical treatments, the lack of tissue targeting of these perturbations makes some of the mechanistic statements too strong. While I appreciate both the excitement and the limitations that working with an emerging model can have, I believe that some of the conclusions must be strengthened before the manuscript can become suitable for publication. I list my major concerns and offer suggestions below.

1) While the images show the D-V alignment of microtubules in the epidermis and amnion, the requirement of oriented microtubule organization specifically in the amnion and the epidermis (rather than just an intact microtubule cytoskeleton everywhere) is not borne out from the drug treatments. “In contrast, dorsal closure arrests during serosa ingression (red arrowhead in Figure 4') due to depolymerized microtubules in both the amnion and the epidermis (blue and white arrowheads, respectively in Figure 4—figure – supplement 3C) impairing epithelial seaming and aborting dorsal closure (red curve in Figure 4)”. Does colcemid treatment also affect the microtubule network in the serosa? Does colcemid affect serosal ingression? Can the authors attempt to locally perturb the microtubule cytoskeleton using caged compounds?

2) The causal relationship between serosal ingression and the fusion of the amnion is not clear. Is the former a prerequisite for the latter? Does the serosa induce microtubule organization in the amnion, through its contractility? Does the ROCK inhibitor affect it? (see “opposing amniotic seams fuse at the midline upon serosa ingression” and “a functional serosa induces proper microtubule alignment”; Figure 4—figure supplement 2')? Also does serosal ingression drive amnion cell elongation? Can the authors laser ablate the serosa during ingression to validate this?

3) Is serosal ingression mediated by apical constriction: does the apical area of serosal cells reduce with time?

4) Is amniotic fusion a prerequisite for epidermal fusion? (see “both processes are necessary for completion of epidermal seaming and dorsal closure”). Again, can the authors ablate the amniotic fusing front and look at the effects on the epidermal seam?

5) The accumulation of F-actin at the point of fusion of the amniotic flanks needs to be better illustrated. Can the authors use the FM dye with phalloidin so that the cell membranes are also labeled?

6) The work will benefit from better discussion and comparisons with work in *Drosophila* and *Tribolium* and other *Megaselia* work on dorsal closure.

*Reviewer #2:*

This paper addresses the evolution of extraembryonic development and dorsal closure in Diptera. The scuttle fly *Megaselia* represents an interesting intermediate between more basal insects with two extensive extraembryonic tissues, amnion and serosa, and *Drosophila* with a single amnioserosa. In *Megaselia* both extraembryonic tissues are still present. However, the amnion appears to be reduced to a single row of cells connected to the dorsal epidermis. One can imagine how the loss of this simple amnion led to a system with only one extraembryonic tissue like in *Drosophila*. Thus, the paper deals with the fascinating question of morphogenetic simplification during evolution. It uses cutting edge fluorescent stainings, live imaging, gene knockdowns and drug treatments to describe and functionally dissect dorsal closure in *Megaselia* and closely compares the findings with *Drosophila*. Overall this is very interesting work. However, I am worried about the variable quality of the data and some of the conclusions. Here are my specific concerns and questions:

General mechanistic questions:

(The Serosa) “is attached to the amnion which in turn connects to the epidermis […] …generates a physical obstruction.”

From the presented data it is not really clear how this connection works so that it can generate a physical obstruction. In Figure 1 the amnion cells appear rather loosely apposed to the serosa prior to rupture. After rupture and during dorsal closure (Figure 1) the serosa definitely has to be connected (or reconnected) to the serosa. The scheme appears to suggest substantial changes regarding the interactions between serosa and amnion during rupture and dorsal closure.

“Amniotic seaming […] is very similar to epidermal seaming…”

Most of the presented data support dissimilarity: the morphology looks different; there is no very sharp boundary between amnion and serosa. Indeed, in most figures the border is rather hard to see. There is no clear actomyosin cable, no *dpp* expression. What you call "seaming" could just be the trace of high level of F-actin left behind by the last serosa cells that are ingressing.

Particular points regarding data quality and presentation:

Figure 1: What does interspersed nuclear architecture mean? The false coloring in Figure 1—figure supplement 1 rather relies on the position of the nuclei, at least as far as I can judge form the presented micrographs.

Figure 1—figure supplement 1. The ISHs have very low quality. B (early *Mab_zen*) could be an over-stained embryo. Amnion-specific expression of *Mab_pnr* in C is not clearly visible. In *Drosophila* (and *Tribolium) pnr* also marks the dorsal epidermis. This could be the same in *Megaselia*.

Figure 1´: A cross section with membrane markers should be added which shows the cellular arrangements of dorsal ectoderm, amnion and serosa giving rise to the scheme B.

Figure 1: What is the identity of the large elongated cells at the dorsal center?

Figure 2: This is a very bad high contrast micrograph taken with completely different microscope setting than the wt control.

Figure 2: The morphological basis for the yellow lines is not visible in the micrographs.

Figure 4´: Transvers optical sections of the embryo shown in 4C should be provided to support schematic sections shown in 4F and F´. How much cell death is occurring during these stages? This can be tested with a number of simple AB stainings.

*Reviewer #3:*

The manuscript by Fraire-Zamora and colleagues investigates the process of dorsal closure in the scuttle fly an insect with a separate amnion and serosa tissues (as opposed to *Drosophila*, which has single amnioserosa tissue). They find that the signaling and some of the morphogenetic processes are similar to *Drosophila* and suggest that the evolution of this process involved changes in tissue number, rather than changes in signaling.

1) I think the manuscript can improve the explanation as to why this finding is interesting. The authors say the "evolution of morphogenesis is generally associated with changes in genetic regulation", but they do not cite evidence for this and so there is little context to assess their claim.

2) There are contradictions to past work, one being a recent paper from the authors of this study (Saias et al., 2015). The authors state, "a *JNK*/Dpp-dependent contractile epidermal cable and the contraction of the amnioserosa tissue power the progression of dorsal closure, where the actomyosin cytoskeleton is essential in force generation". In Saias et al. (2015), they argued that myosin antagonized closure, this is confusing.

3) While the "seaming" is interesting, dorsal closure can occur without the seams. See Wells et al., 2014. The authors need to cite this paper and rationalize why they focus on the "seaming" process, which does not seem to be the major force generator.

[Editors’ note: what now follows is the decision letter after the authors resubmitted.]

Thank you for resubmitting your work entitled "Two consecutive microtubule-based epithelial seaming events mediate dorsal closure in the scuttle fly *Megaselia abdita*" for consideration by *eLife*. Your article has been evaluated by a Senior Editor and three peer reviewers, one of whom is a member of our Board of Reviewing Editors. The reviewers have opted to remain anonymous.

Our decision has been reached after consultation between the reviewers. Based on these discussions and the individual reviews below, we regret to inform you that your work will not be considered further for publication in *eLife*.

The revised version of the manuscript by Solon and colleagues was reviewed by two of the reviewers of the original manuscript and a third new reviewer. As you will see from their individual comments below, all reviewers agree that the question that the paper addresses is very interesting. The two original reviewers also appreciate the effort that the authors have made to address some of the queries raised by the reviewers in the previous version (including orthogonal views, additional images and analysis to better define tissue topology and new laser ablation experiments to test the requirement for the cellular behaviours such as serosal ingression and the two seaming events in the process). All three reviewers however feel that the description of the process and the cellular and subcellular mechanisms invoked as being important are not sufficiently strongly backed up by the data presented whose quality is also variable. The new experiments are also not conclusive in their present form and are not critically evaluated by the authors. In addition, the new reviewer makes several very important points about the methodologies used and suggests alternative interpretations of the data presented (the *zen* RNAi experiments used to establish the critical requirement of the serosa, the "fast" dynamics of the amnion during "seaming" and the non-specific/global effects of microtubule inhibitors).

Collectively, all three reviewers suggest that in addition to toning down the strength of the statements that imply mechanism, the authors must work on generating high quality data to strengthen the claims made about the cell biology of the process, taking into account several useful points made by the reviewers, for future submissions.

*Reviewer #1:*

In their revised manuscript, Solon and colleagues have attempted to address some of the queries raised by the reviewers in the previous version. Notably, they present some orthogonal views and additional images and analysis to better define tissue topology and perform new laser ablation experiments to test the requirement for the cellular behaviours (serosal ingression and the two seaming events) in the process. I appreciate the challenges and excitement of studying a new model organism, and that it is not trivial to obtain mechanistic cellular and molecular insights. I also appreciate the effort the authors have made to address the questions raised. I still feel that the description of the process and the cellular and subcellular mechanisms invoked as being important are not sufficiently strongly backed up by the data presented. Without a clear description of the topology of the process (even from fixed preparations with good orthogonal views), it is hard to interpret the consequences of the drugs used to perturb actomyosin contractility and the microtubule network, both of which appear to have widespread effects (including the patterning of and shape changes in the epidermis). Also, it is not possible to conclude much from the laser ablation experiments as presented here, which are very qualitative, other than that in some instances, it produced recoil and the process failed. I therefore feel that the statements made in the paper that relate to i) physical interactions between the participating tissues, ii) requirement of actomyosin contractility and iii) the two microtubule rearrangements for specific cell behaviours during dorsal closure are still too strong. I think the quantitative description of the tissue dynamics presented here (serosal area and height dynamics), if combined with a good cell biological description of the tissue topologies, interactions and cell behaviours and cytoskeletal organization, will in itself constitute an important piece of work in a new model organism and getting better quality data to strengthen these aspects will add significantly to our understanding of the conservation of developmental mechanisms.

I list my detailed comments below:

1) While it is clear from the videos that serosal rupture and contraction (reduction in area of the sheet) occur, it is still not clear to me what the authors mean by serosal ingression. Looking at the videos, it appears as if the serosa ruptures and retracts, collapses over the dorsal opening and then degenerates. While the schematics are useful, the data presented do not convincingly reveal the earlier close apposition of the serosa and the vitelline membrane, serosal ingression, or the subsequent reapposition of the serosa and vitelline membrane (as depicted in the schematics in Figure 1). What would be very useful, as suggested earlier, is a staged series of images of confocal projections in a lateral view, and dorsal-up views with their orthogonal sections. Phalloidin and DAPI should reveal these topologies and, at least in *Drosophila*, some phalloidin can get in through the vitelline membrane.

2) The authors measure the areas of serosal cells at different stages during dorsal closure and show that the apical areas reduce over time. This addresses a query I had raised in the previous version. The authors also show that the area of the serosa rapidly reduces. It would be desirable to see in an orthogonal view, what happens to the apicobasal axis of this sheet during contraction and ingression.

3) The correlation between cell elongation and microtubule organization suggests that the latter follows the former and may be a consequence of shape change (subsection “A microtubule-based seaming of the extraembryonic amnion is required for dorsal closure in *Megaselia abdita* embryos”, first and fourth paragraphs). The colcemid experiments indicate that the microtubule cytoskeleton is necessary for the timely completion of closure but do not point specifically to a role for microtubule rearrangements in the seaming events described.

4) The effects of both the ROCK inhibitor (subsection “A microtubule-based seaming of the extraembryonic amnion is required for dorsal closure in *Megaselia abdita* embryos”, second paragraph, Video 5) and colcemid (Video 10) appear to be widespread. Specifically they do not help in resolving causality between cytoskeletal organization and cell behaviour. Both also appear to alter cell shapes/ pattern in much of the epidermis, which may also contribute to the effects on dorsal closure.

5) The statements made about attachment between the tissues (Figure 2) are too strong and the schematics are not backed up by the data in the images. It is probably okay to say close apposition but active attachment is not reflected by the data shown (subsection “Dorsal closure in *Megaselia abdita* involves synchronized serosa rupture and epidermal progression”, third paragraph).

6) In the laser ablation experiments, the ROI ablated should be marked, the power regimes used and the number of embryos ablated for each experiment should be mentioned in methods. While some experiments show tissue recoil, the effect of ablation on the completion of dorsal closure is less clear. Overall, these experiments appear to be very qualitative.

*Reviewer #2:*

The authors have comprehensively responded to the reviewers’ concerns. They added new stainings and analyzed cross section to clarify the tissue topology. They also conducted new functional experiments; in particular, they performed laser ablations of serosa and amnion cells followed by live imaging. Most of the new material is convincing and supports the main conclusion of the paper that two subsequent epithelial fusion events are required for dorsal closure in *Megaselia*. I have however still a serious problem with variable quality of figures and videos.

Here is a list of figures and videos of low quality:

Figure 1—figure supplement 1

B, B´ I would remove this *zen* ISH. Both could be artefacts. This result is not important for the paper.

D plus Video 1: it is not clear what one can learn from this video. I suggest removing.

It is very laudable that the authors tried to do laser ablations of amnion and serosa cells. However, the video and stills from these ablations are very hard to interpret. The side of ablation and the cellular consequences are hardly visible. I am not able to interpret the pictures shown in Figure 4—figure supplement 3 and Video 8 does not help. The same applies for the amniotic ablations shown in Figure 4—figure supplement 5. Indeed, I found none of the laser ablation movies (Video 6–Video 8) informative.

It is also laudable that the author tried to produce optical cross section which in many cases helped to better understand tissue arrangements. However, the cross sections shown in Figure 4—figure supplement 6 seem not to be informative to me.

Taken together the paper has clearly improved. But the authors unfortunately added material which does not really show what they want to say. I think most of this material can be just removed.

*Reviewer #4:*

The manuscript by Fraire-Zamora and coworkers deals with cell biological processes underlying dorsal closure in *Megaselia abdita*, a fly species that develops two distinct extraembryonic tissues, the amnion and serosa. The findings in *Megaselia* are placed into an evolutionary context by comparison with extraembryonic development and dorsal closure in *Drosophila melanogaster*, which develops only a single extraembryonic tissue, the amnioserosa. This comparison is aimed to gain insights into cell biological changes underlying the evolutionary transition in extraembryonic tissue development and topology. The question is very exciting as it aims to addresses how, in addition to changes in gene regulation and signaling, changes in epithelial organization and cell behavior contributed to the origin of evolutionary novelty.

The authors describe dorsal closure in *Megaselia* as two subsequent events. They suggest that, first, the serosa is internalized and the amnion is seamed along the dorsal midline. Then, the amnion is internalized and the ectoderm is seamed along the dorsal midline. Both seaming events are described to be dependent on microtubule alignment, and microtubule alignment in the amnion (and, as a consequence, dorsal closure) is suggested to be dependent on a functional serosa. As a result, the two extraembryonic tissues may be seen as having the capacity to self-coordinate dorsal closure by a different cell biological mechanism than *Drosophila* despite conservation of late *JNK*/Dpp signaling.

While I find the manuscript conceptually intriguing, I am not sure that all major conclusions are sufficiently supported by the presented data. In addition, I have some issues with methods and data quality.

Comments on major conclusions:

1) The conclusion that the serosa is critical for alignment of the microtubule cytoskeleton is based on the analysis of genetic serosa depletion in *zen* RNAi embryos, the pharmacological inhibition of serosa cell activity by injecting ROCK inhibitor Y-27632, and by physical laser ablation of the internalizing serosa. Laser ablation and pharmacological inhibition of serosa cell activity leave behind tissue that can interfere with the behavior of the amnion. Thus the genetic depletion of the serosa in *zen* RNAi embryos represents a key experiment for the above conclusion.

In the analysis of their *zen* RNAi results, the authors report early arrested embryos. Because this includes embryos dying for unspecific reasons (e.g. by the injection procedure per se), I would suggest to omit this data. The adjusted data shows that 2/3 of *zen* RNAi embryos develop a dorsal open phenotype and 1/3 of *zen* RNAi embryos show wildtype development. This is in line with previous results for *Megaselia* and *Tribolium*, which have shown that dorsal closure occurs in about 50%-75% of *zen/zen1* RNAi embryos (Rafiqi et al., 2008; van der Zee et al. Current Biology 15, 624; Panfilio, Oberhofer and Roth, 2013). Discrepancies in the proportion of dorsally closed embryos in *zen* RNAi embryos reported here and in previous publications may stem from a method modification: because the authors fix the embryos prior to cuticle preparation, and because embryonic development is typically delayed after dsRNA treatment, it is possible that some embryos were fixed before cuticle formation was completed and thus were mistakenly classified as dorsal open.

The authors show that in *zen* RNAi embryos microtubules are no longer aligned as in wildtype (Figure 4—figure – supplement 2B'), yet dorsal closure still occurs in 1/3 of *zen* RNAi embryos. The authors suggest that dorsal closure in *zen* RNAi embryos is due to incomplete gene knockdown, the development of a remnant serosa, and a local alignment of microtubules close to the remnant serosa. The key observation here is an accumulation of local F-actin, which the authors interpret as remnants of serosal tissue [subsection “A microtubule-based seaming of the extraembryonic amnion is required for dorsal closure in *Megaselia abdita* embryos”, third paragraph]. I do not think that local actin accumulation is sufficient evidence to interpret cells as remnants of serosal tissue, in particular since main conclusions of the paper depend on this result (1. microtubules need serosa to align, 2. aligned microtubules are required for dorsal closure). To support their interpretation convincingly, I would suggest the authors build on their defined tissue-specific features of the serosa [subsection “Dorsal closure in *Megaselia abdita* involves synchronized serosa rupture and epidermal progression”, first paragraph], and use them to demonstrate the presence of serosal remnants in *zen* RNAi embryos by marker gene expression (*zen, ddc* could be an alternative (Rafiqi et al., 2010)), nuclear size, and DNA compactness.

2) The conclusion that the amnion fuses by seaming along the dorsal midline is based on time lapse recordings after FM4-64 injection (Video 9). The video shows how the flanking cells of the amnion meet along the dorsal midline after the serosa is internalized. Following serosa internalization and amnion touching, the ectoderm meets along the dorsal midline. The two processes seem to follow very different dynamics. In case of amnion cell meeting along the dorsal midline, cells touch and close a gap over the serosa at a speed of about 6 µm/min (*i.e.* 30 µm at 1:10-1:15). In case of the ectoderm cell meeting along the dorsal midline, cells touch and close a gap over the amnion at a speed of about 1.2 µm/min (*i.e.* 30 µm at 1:50-2:15). While the dynamics in the slow fusion in the ectoderm appear consistent with the idea of seaming, I am concerned that the fast touching of the amnion cells is not an equivalent process. Rather than a seam of fused cells, the somewhat brighter cell boundary between touching amnion cells along the dorsal midline may just reflect the increased intensity due to the touching membranes of two amnion cells and/or remnants of actomyosin accumulation at the serosa/amnion interface.

3) The conclusion that microtubule organization in the amnion is required for epithelial seaming and dorsal closure is based on time-lapse recordings after injection of colcemid, which leads to what is described as "dorsal closure arrests during the initial stages of serosa ingression". I am not quite sure the Video shows an "arrest". When serosa internalization does not proceed further (02:10), dorsal closure seems to reverse rather than being arrested. I am worried that instead of showing a specific effect of microtubule depolymerization, this is due to a collapse of the system and dying of the embryo. Because this is a key experiment aimed to demonstrate a role of microtubules in dorsal closure, it would be necessary to address such concerns by providing more information (how often has this phenomenon been observed in independent experiments, what are possible explanations for the observed serosa eruption) and/or by a more local perturbation of microtubule organization in the amnion.

4) The conclusion that amnion seaming is a prerequisite for epidermal seaming is based on laser-ablation. Following laser ablation of what is described to be the amniotic-seaming front, epidermal seaming stops and results in a late dorsal open phenotype. It is difficult to follow the experiment based on the provided time-lapse recordings. In the wildtype reference (Video 9), ectoderm, amnion, and serosa cells can be clearly distinguished. In the time-lapse recording that is meant to show ablation of the amniotic-seaming front (Video 11), neither amnion nor serosa cell outlines are visible. Thus it is not possible to localize the laser ablation and what tissue has been affected. The visualization of the amnion, the area in which the amnion cells touch along the dorsal midline, and the ablation precisely where amnion cells touch, is, however, critical to demonstrate that dorsal closure is impaired due to interference with "amnion seaming" (see above for concerns about "seaming") and not just due to the loss of amnion tissue integrity.

Comments on data quality:

1) The authors use nuclear size and DAPI density to characterize tissue type [subsection “Dorsal closure in *Megaselia abdita* involves synchronized serosa rupture and epidermal progression”, first paragraph]. How was area and density of DNA stain in *Megaselia* defined as faithful descriptors of cell type? In its current presentation, the coloration of nuclei appears a bit arbitrary, even though it is most likely based on a double staining of marker genes (*zen* for serosa, *pnr* for amnion) and DNA stain to correlate nuclear size and DNA density with tissue types. To explain their rationale better, the authors should add an image of a such double staining and illustrate how the quantitative and tissue-specific parameters of nuclear size and "DNA compactness" were obtained.

2) Classification of serosa and amnion in Figure 1—figure – supplement 1A is puzzling. The red nucleus on top of the third blue nucleus from the right seems to have a size that fits into the amnion category. What is the basis to classify this nucleus as serosal? Is it the density of the DNA stain? It does not become clear by which procedure two qualitatively different parameters (nuclear size and "DNA compactness") were compared if they contradict each other in the classification of a particular nucleus. Probably this is done by some sort of quantitative measure – the authors should comment on this procedure in more detail in the Materials and methods section.

3) The density of serosa cells in Figure 1—figure – supplement 1A – are unexpectedly variable, it seems to be much higher in areas where the serosa covers the ectoderm than in areas covering the interface of amnion and ectoderm, the amnion, or the yolk. In previous publications and other images within this publication (e.g. Figure 1, Figure 1—figure – supplement 2A), the density of serosa nuclei appears to be rather even along the dorsal to ventral circumference of the embryo. Could the authors offer an explanation for this discrepancy?

4) Whole mount in situs of *Mab_zen* and *Mab_pnr* are not of high quality. In situs in *Megaselia* should be an established procedure, also for embryos at late stages (e.g. Rafiqi et al., 2010). In their description of gene expressions, the authors do not mention the staining of *Mab_zen* and *Mab_pnr* probe in the head and *Mab_zen* probe along the ventral midline. Do they have reasons not to trust this aspect of the staining? Dorsal staining of *Mab_zen* probe and *Mab_pnr* probe could be genuine, but this is difficult to evaluate at this quality level: poorly devitellinized embryos can lead to similar dorsal staining through unspecific probe accumulation at cracks in the vitelline membrane.

5) The authors suggest that devitellinization results in damage of the dorsal region of the embryo [subsection “Dorsal closure in *Megaselia abdita* involves synchronized serosa rupture and epidermal progression”, third paragraph], where they do not detect phalloidin staining (Figure 1—figure – supplement 2B). This suggest the authors have evidence that the amnion is normally closed at this point of development. This is contradicting the schematic overview in Figure 1'. To clarify, it would be helpful to provide a brief description and maybe a panel with a closed amnion to illustrate the progression of wildtype amnion development.

6) What is the stage of dorsal closure in Figure 1—figure – supplement 2D-D'? In the schematic overview (Figure 1—figure – supplement 2E) it seems as if the serosa still envelopes most part of the embryo. The embryo shown in 2D-D' seems to be older and serosa internalization well under way, corresponding to the schematic overview in Figure 1'. The authors explain that serosa tissue can be lost during devitellinization, suggesting that they used alternative landmarks to stage dorsal closure progression. It would be helpful if they could comment on their use of landmarks and how the embryo in 2D-D' was staged.

7) The authors suggest that amnion and serosa share a junction or attachment site [subsection “Dorsal closure in *Megaselia abdita* involves synchronized serosa rupture and epidermal progression”, third paragraph]. The increase of phalloidin staining in Figure 1—figure – supplement 2D' is sufficient evidence that the cell with blue arrow touches the cell with white arrow. A blob of actin does not provide evidence for junctions, and the black lines indicating amnion-serosa attachment in schematic in Figure 1—figure – supplement 2E may be interpreting the available data a bit too much.

[Editors’ note: what now follows is the decision letter after the authors submitted for further consideration.]

Thank you for resubmitting your work entitled "Two consecutive microtubule-based epithelial seaming events mediate dorsal closure in the scuttle fly *Megaselia abdita*" for further consideration at *eLife*. Your revised article has been favorably evaluated by K VijayRaghavan (Senior editor) and two reviewers, one of whom is a member of our Board of Reviewing Editors.

The manuscript has been improved but there are some remaining issues that need to be addressed before acceptance, as outlined below:

Both reviewers agree that the revised version is a substantial improvement over the earlier versions both with respect to the conclusions and the readability. Specifically, the new data presented clarifies many issues on the topology of interacting tissues and the cell shape changes accompanying serosal contraction and provides a better resolution of the two seaming events. The reviewers were also impressed with the experiments designed to locally inactivate colcemid. Both reviewers had a few concerns that must be addressed (in a revision) and made the following suggestions.

i) While the orthogonal sections presented do a good job of showing tissue topology, serosal ingression and its accompanying cell shape changes, the final configuration of the three tissues is not clear. It should be possible to resolve this by making orthogonal optical sections at different locations along the AP axis (at the canthus and closer to the AP midline) from the images presented in Figure 1—figure supplement 1'", phalloidin DAPI and Figure 4—figure supplement 1"', tubulin, DAPI, since closure proceeds inwards from the canthi.

ii) The schematics depicting the changes in the three tissues participating in dorsal closure must be modified to accurately depict the serosal cell shape changes described and to indicate the final configurations of the three tissues at the end of the closure.

iii) The data on the local colcemid inactivation includes two images and quantification of dorsal closure height. The authors must state when colcemid was injected and when it was inactivated and describe in Materials and methods how the ROI was created. It would be desirable to include the video of the treatment.

iv) Could the authors comment on the fate of the amnion: What happens upon fusion: does it detach from the epidermis? Or does it ingress with the serosa? Again orthogonal optical sections from the fused regions should help resolve this.

v) Can the authors clarify if they are suggesting that the "movement " of amnion cells (devoid of actomyosin) is microtubule dependent and whether serosal contraction contributes to bringing the amnion flanks into close proximity?

---

## [Author Response]

[Editors’ note: the author responses to the first round of peer review follow.]

Reviewer #1:

[…] 1) While the images show the D-V alignment of microtubules in the epidermis and amnion, the requirement of oriented microtubule organization specifically in the amnion and the epidermis (rather than just an intact microtubule cytoskeleton everywhere) is not borne out from the drug treatments. “In contrast, dorsal closure arrests during serosa ingression (red arrowhead in Figure 4') due to depolymerized microtubules in both the amnion and the epidermis (blue and white arrowheads, respectively in Figure 4—figure supplement 3) impairing epithelial seaming and aborting dorsal closure (red curve in Figure 4)”. Does colcemid treatment also affect the microtubule network in the serosa? Does colcemid affect serosal ingression? Can the authors attempt to locally perturb the microtubule cytoskeleton using caged compounds?

The reviewer wonders about the interpretation of the colcemid experiments and whether contribution from defects in serosa retraction and ingression could also contribute to the observe phenotype. We have performed additional immunostaining of colcemid treated embryos that show that the microtubule network in the serosa is also altered. These images are now in Figure 4—figure supplement 4’. In order to verify whether colcemid treatment would affect serosal retraction and ingression, we have quantified the kinetics of retraction in these embryos and it appears unchanged compared to wild-type. This quantification now appears in a revised Figure 4—figure supplement 4 of the manuscript. Serosa ingression in injected embryos does not progress, an expected outcome in the absence of seaming. For instance, in *Drosophila*, the last stages of amnioserosa contraction are also impaired with colcemid injection and the absence of epidermal seaming (Jankovics and Brunner 2006). Regarding the use of caged compounds, this would be a wonderful experiment to perform. However, this is challenging in a non-model organism where fluorescent labels are generally weak. Additional use of a UV laser may render the signal even weaker. Furthermore, to our knowledge, such compounds that perturb the microtubule cytoskeleton are not available commercially.

2) The causal relationship between serosal ingression and the fusion of the amnion is not clear. Is the former a prerequisite for the latter? Does the serosa induce microtubule organization in the amnion, through its contractility? Does the ROCK inhibitor affect it? (see “opposing amniotic seams fuse at the midline upon serosa ingression” and “a functional serosa induces proper microtubule alignment”; Figure 4—figure supplement 2')? Also does serosal ingression drive amnion cell elongation? Can the authors laser ablate the serosa during ingression to validate this?

To assess the influence of the serosa on microtubule organization and amnion cell elongation, we have performed immunostaining of microtubules and actin in contractility-impaired embryos (injected with Rock inhibitor Y27632). In these images, we did not observe preferential orientation of the microtubule network or elongation of the amnion cells towards the serosa. These figures are added in the new Figure 4—figure supplement 3’.

In addition to Y-27632 treatment, we have performed laser dissection experiments to ablate the serosa. These experiments are challenging because embryo labeling with FM 4-64 is not optimal to follow individual cell behavior in our laser dissection system. However, we managed to ablate the serosa and imaged ablated tissue in a different microscope with better resolution (see Materials and methods). We succeeded to observe that, upon serosa ablation, amnion cells adjacent to the ablated area do not elongate towards the dorsal midline (along the dorsoventral axis). Instead, amnion cell shape relaxes and follows the edge of the ablated serosa. Also, upon ablation and epidermal edge advancement, the shape of intact amnion cells changes progressively towards an anteroposterior elongation. This suggests that forces generated by the serosa contribute to amnion cell elongation along the dorsoventral axis. When these forces are not present, amnion cells fail to elongate dorsoventrally and instead elongate in an anterio-posterior fashion. The final result of serosa ablation is failure of dorsal closure as observed by a dorsal-open cuticle phenotype. The results of these laser-ablation experiments are now in the new Figure 4—figure supplement 3, Video 6 and Video 8.

3) Is serosal ingression mediated by apical constriction: does the apical area of serosal cells reduce with time?

We have measured serosa cell size over time in FM 4-64-labeled embryos and observed a reduction in cell size from ~ 200 µm^2^ to approximately 30 µm^2^. In addition, we have assessed actin accumulation at the most apical sites of the ingressing serosa cells in fixed embryos. It is difficult to precisely locate the apical region in *M. abdita* embryos as FM 4-64 labels both apical and basal membranes, and we do not have antibodies that are specific for apical markers in this species. Both, actin accumulation at apical sites of the ingressing serosa and cell size reduction over time are displayed in the revised Figure 3—figure supplement 1. The observed apical accumulation of actin is consistent with apical constriction, and therefore we have modified the text as follows:

“Actin accumulates apically in contracting serosa cells (red arrowheads in Figure 3—figure supplement 1). This accumulation correlates with cell size reduction over time (Figure 3—figure supplement 1), suggesting an apical constriction mechanism during serosa ingression.”

4) Is amniotic fusion a prerequisite for epidermal fusion? (see “both processes are necessary for completion of epidermal seaming and dorsal closure”). Again, can the authors ablate the amniotic fusing front and look at the effects on the epidermal seam?

We have used laser dissection to ablate the amniotic seaming front and followed epidermal seaming after ablation (see Materials and methods). We observed that epidermal fusion progresses at sites where amnion cells remained intact, until the epidermal front reached the ablated amnion site. At this point, epidermal progression is arrested. Ablation of the amniotic seeming front resulted in incomplete epidermal closure and a small gap on the dorsal side (that we define as a late dorsal-open phenotype). These data indicate that amniotic fusion is a prerequisite for epidermal fusion. They are shown in the new Figure 4—figure supplement 5 and Video 7 and 11.

5) The accumulation of F-actin at the point of fusion of the amniotic flanks needs to be better illustrated. Can the authors use the FM dye with phalloidin so that the cell membranes are also labeled?

We have attempted to co-label membrane and phalloidin with FM 4-64 dye. However, we were not able to resolve membranes properly in fixed embryos (see Author response image 1). Instead, we have incorporated an image where the accumulation of actin along the site of amniotic fusion (white arrowhead in a and b) is clearer. By performing an orthogonal stack reslice (b), we observed that this accumulation is at the same z-plane as the epidermal actin cable (yellow arrowheads). Both images are included in Figure 3’ (without the FM 4-64 label). The actin accumulation at the site of fusion also colocalizes with the FM 4-64 signal that we observe in fixed embryos, however, we do not incorporate this image since we think the resolution of membranes is not good enough for publication purposes.

6) The work will benefit from better discussion and comparisons with work in Drosophila and Tribolium and other Megaselia work on dorsal closure.

We have modified the text to include comparisons with *Tribolium, Drosophila* and previous *Megaselia* work on dorsal closure.

Reviewer #2:

[…] General mechanistic questions:(The Serosa) “is attached to the amnion which in turn connects to the epidermis […] generates a physical obstruction.”From the presented data it is not really clear how this connection works so that it can generate a physical obstruction. In Figure 1 the amnion cells appear rather loosely apposed to the serosa prior to rupture. After rupture and during dorsal closure (Figure 1) the serosa definitely has to be connected (or reconnected) to the serosa. The scheme appears to suggest substantial changes regarding the interactions between serosa and amnion during rupture and dorsal closure.

We have performed additional staining both in live-intact embryos (DAPI injection) and in fixed devitellinized embryos. The resulting images provide better evidence on the anatomy of the three-tissue system and confirm the connection between the serosa, amnion and epidermis. These images are in the new Figure 1—figure supplement 2. We have modified the schemes in Figure 1’ and 1G, G’ and G’’ to illustrate our observations in a more parsimonious way. We also added a scheme in Figure 1—figure supplement 2 of the physical obstruction that the serosa-amnion junction represents for the advancement of the epidermal front.

“Amniotic seaming […] is very similar to epidermal seaming…”Most of the presented data support dissimilarity: the morphology looks different; there is no very sharp boundary between amnion and serosa. Indeed, in most figures the border is rather hard to see. There is no clear actomyosin cable, no dpp expression. What you call "seaming" could just be the trace of high level of F-actin left behind by the last serosa cells that are ingressing.

We have modified the text to better describe our observations:

“Our experimental data indicate that amniotic seaming is microtubuledependent and is essential for dorsal closure to occur. Thus, similar to epidermal seaming in *D. melanogaster* embryos where microtubules align dorsoventrally prior to tissue fusion (Jankovics and Brunner, 2006), amniotic and epidermal seaming in *M. abdita* also rely on dorsoventral alignment of microtubules. This suggests that microtubule alignment may be a general hallmark for fusing epithelia.”

It is difficult to assess whether F-actin accumulation at the merging site is exclusively left by the last ingressing serosa cell or exclusively from the amnion seaming process. The lack of UAS-GAL4 lines for *M. abdita* makes difficult to target specific tissues to obtain high-resolution data of whether Factin accumulation is exclusively amniotic.

As explained above to reviewer 1, we have observed that F-actin accumulation is within the same confocal plane as the epidermal actin cable and is stained with the FM 4-64 dye. We have modified the text in the manuscript to state this observation in a clear way and avoid misinterpretations:

“upon serosa ingression, the amniotic flanks (devoid of actomyosin) move to the dorsal midline where F-actin accumulates at the site where the two flanks merge and serosa ingresses (white arrowhead in Figure 3’’ and 3D). This Factin accumulation is continuous along the merging site and localized in the same confocal plane than the actomyosin cable surrounding the amnion (white arrowhead in Figure 3’).”

Particular points regarding data quality and presentation:Figure 1: What does interspersed nuclear architecture mean? The false coloring in Figure 1—figure supplement 1 rather relies on the position of the nuclei, at least as far as I can judge form the presented micrographs.

We have modified the text to describe nuclear staining rather than nuclear architecture. The false coloring is based on the area of the different nuclei, indicating different populations of cells. We have included the areas in the text, added a reference that used nuclear anatomy and staining to identify extraembryonic tissue in *Tribolium* (Panfilio et al., 2013) and stated that cellular identity is based on nuclear size, position and staining with DAPI and ISH markers (*Mab_zen* for serosa and *Mab_pnr* for amnion).

Figure 1—figure supplement 1. The ISHs have very low quality. B (early Mab_zen) could be an over-stained embryo. Amnion-specific expression of Mab_pnr in C is not clearly visible. In Drosophila (and Tribolium) pnr also marks the dorsal epidermis. This could be the same in Megaselia.

We have substituted the *Mab_zen* image for a new one in which the ISH stains the nuclei in some ruptured serosa cells, revealing the underneath unstained epidermis. We would like to point out that obtaining a good ISH of the serosa covering the embryo is difficult because, during heat fixation and manipulation of the embryo, the serosa is mostly removed together with the vitelline envelope to which it adheres.

Regarding amnion markers, in *M. abdita*, there is not a specific marker for the entire amnion (Rafiqi et al., 2010). Thus, different markers have been used to identify extraembryonic amniotic tissue. These *M. abdita* amnion markers are: C15, *Krüppel (Kr), pannier (pnr*) and *hindsight (hnt). hnt* seems to be more specific for amnion at late stages (despite being also expressed in the embryonic epidermis), as it has been observed in amniotic extraembryonic tissue during early germ band retraction (Rafiqi et al., 2008, 2010 and 2012). In our hands, ISH of *Mab_hnt* in embryos at dorsal closure stage does not stain extraembryonic tissue (see Author response image 2). Our *Mab_hnt* probe stains gut (a) and epidermis (b), as expected, but is not visible in the amnion (blue arrowhead in c).

**Author response image 2. respfig2:** 

Since C15 also stains serosa and epidermal edge cells and *Kr* also stains developing serosa, we used *pnr* as an amnion marker despite it also marking embryonic epidermis. We have reproduced ISHs of *pnr* that show a faint staining in the amnion and parts of the epidermis (see red arrowheads in Author response image 3).

**Author response image 3. respfig3:** 

We have modified the text to clarify that we use nuclear size and staining together with *Mab_pnr* to identify amnion cells and differentiate them from serosa and epidermis:

“The extraembryonic amnion, which is 1-2 cell wide, localizes in between the serosal and epidermal tissues (blue in Figure 1’ and 1B). Its cells have large nuclei (average size 55 *±* 13 µm^2^, SD, *n*= 10 cells), show a continuous, “compact” DAPI staining (blue in Figure 1’ and Figure 1—figure supplement 1’) and express the amnion marker gene *pannier (pnr*) (black arrowheads in supplementary Figure 1).”

Figure 1´: A cross section with membrane markers should be added which shows the cellular arrangements of dorsal ectoderm, amnion and serosa giving rise to the scheme B.

As mentioned above to reviewer 1, plasma membrane imaging in fixed embryo is challenging, even more for non-devitellinized intact embryos that will preserve the original three-tissue system anatomy. FM 4-64 dye in fixed embryos does not provide a clear staining. We injected different membrane dyes (BODIPY, Dil, DiO and FM 4-64), but only FM 4-64 diffuses along the whole embryo surface. Alternatively, we tried to generate cryosections of *M. abdita* embryos stained with DAPI and methylene blue and embedded in OCT compound (see Author response image 4, dorsal is to the top and ventral to the bottom) as described in Panfilio et al., 2006 for *Oncopeltus fasciatus*. We were able to visualize serosa cells attached to the vitelline envelope and close to the dorsal part of the embryo (white arrowhead in b), however, the resolution of such sections is not good enough to be conclusive.

**Author response image 4. respfig4:** 

On the other hand, we hand-devitellinized fixed embryos and immunostained for phalloidin and DAPI. In most of the cases, devitellinization also removed the serosa cells that are attached to the vitelline membrane in embryos at dorsal closure stage, and left ruptured amnion and intact epidermis in the embryo. In a couple of embryos, the serosa remained intact on top of the amnion and epidermal cells. An orthogonal stack reslice of the intact serosa on top of amnion and epidermis reveals the anatomy of the *M. abdita* three-tissue system, the relative position of serosa, amnion and epidermis and the shared junction by the serosa and amnion cells. These images are added in the new Figure 1—figure supplement 2.

Figure 1: What is the identity of the large elongated cells at the dorsal center?

The identity of these cells with large elongated nuclei is unclear to us. They appear to be embedded in the yolk. They can be visualized clearly in Figure 1—figure supplement 2 and in Video 1. We have described them in the text as unidentified cells. In *Drosophila*, similar cells are also located within the yolk sac; these cells embedded in the yolk are thought to be haemocytes or crystal cells, however, we have no means to pin down the identity of the large cells with elongated nuclei in the yolk of *M. abdita*. We have described them in the text of the manuscript.

Figure 2: This is a very bad high contrast micrograph taken with completely different microscope setting than the wt control.

We have substituted the images by two micrographs taken using the same microscope and camera settings.

Figure 2: The morphological basis for the yellow lines is not visible in the micrographs.

We have included the image processing method that allowed us to draw the contour of the serosa during retraction in bright-field image sequences. Due to low contrast in these images, we subtracted an image at time *t*+1from the image at time *t*. This simple operation allowed to highlight morphological changes between frames and to detect the reduction in serosa contour. This is included in Figure 2—figure supplement 1 and described in Materials and methods.

Figure 4´: Transvers optical sections of the embryo shown in 4C should be provided to support schematic sections shown in 4F and F´.

We have now added transverse section (obtained by orthogonal stack reslicing) corresponding to Figure 4 in the Figure 4—figure supplement 6A-a’’. In these optical transverse sections we can identify the invagination and ingression of the serosa cells as depicted in the schematics on the Figure 4.

How much cell death is occurring during these stages? This can be tested with a number of simple AB stainings.

We have used some markers for apoptosis to probe for cell death around dorsal closure stage. First, we used an antibody against cleaved caspase-3 that did not show any signal during serosa ingression in *M. abdita* embryos (see Author response image 5’) compared to DAPI and microtubule immunostaining in the same embryo (a and a’’). The same antibody did not show signal during germband retraction (b), early dorsal closure (c) or after epidermal seaming (d). We also stained live *M. abdita* embryos with acridine orange, a dye used to evidence acidification during cell death, and did not observe a strong signal (e), except for some puncta that could be yolk granules or heamocytes. Injection of Annexin V, an early marker of apoptosis, showed a punctuated signal in the serosa (red arrowhead in f) that increased as the serosa retracted (f’) and at the time of ingression (f’’). Annexin V is a Ca^2+^-dependent protein that binds to phosphatidylserine (PS) when this phospholipid is externalized and exposed at the cell surface during early steps of apoptosis. Since Annexin V is not able to penetrate the lipid bilayer, it will only stain cells that are undergoing apoptosis, however, cells that are undergoing plasma membrane damage will allow Annexin V to enter the cell and stain PS in the inner leaflet of the bilayer (Bundscherer et al., 2014). Since serosa cells are also undergoing plasma membrane rupture, we cannot establish whether Annexin V is staining PS from the inner leaflet in damaged cells or externalized PS from early apoptotic events. Thus, we do not have conclusive evidence for cell death during dorsal closure in *M. abdita* embryos. We believe that, even if apoptosis may play an important role during the process, it is beyond the scope of the manuscript.

**Author response image 5. respfig5:** 

Reviewer #3:

[…] *1) I think the manuscript can improve the explanation as to why this finding is interesting. The authors say the "evolution of morphogenesis is generally associated with changes in genetic regulation", but they do not cite evidence for this and so there is little context to assess their claim.*

We have now included in the text references of work proposing that evolution of morphogenesis and development is generally associated with changes in genetic regulation: (Carroll et al., 2009; Davidson and Erwin, 2006; Peter and Davidson, 2015; Wilkins, 2002)

2) There are contradictions to past work, one being a recent paper from the authors of this study (Saias et al., 2015). The authors state, "a JNK/Dpp-dependent contractile epidermal cable and the contraction of the amnioserosa tissue power the progression of dorsal closure, where the actomyosin cytoskeleton is essential in force generation". In Saias et al. (2015), they argued that myosin antagonized closure, this is confusing.

There are no such contradictions. In *Drosophila*, actomyosin contractility is an essential force for closure progression. When amnioserosa cell contractility is suppressed, dorsal closure does not occur (see Scuderi and Letsou 2005, and Caussinus et al. 2012). In Saias et al. 2015, we show that, in addition of the contractility of the AS cells, a force emanates from the decrease in volume of the amnioserosa cells triggered by apoptosis. In Saias et al., we indeed have shown that down-regulation of myosin accelerates the first stages of closure, consistently with our 3D model of the amnioserosa cells where lateral tension would antagonize closure. However, we also clearly stated that myosin contractility is an essential force for closure and that in late dorsal closure stages in *Drosophila*, myosin enrichment is observed at the apical site of the cells.

We have modified the text in the manuscript to clarify this possible misunderstanding:

“In *D. melanogaster* embryos, a *JNK*/Dpp-dependent contractile epidermal cable and the contraction of the amnioserosa tissue (through actomyosin contractility and volume decrease) power the progression of dorsal closure. (Hutson et al., 2003; Kiehart et al., 2000; Saias et al., 2015). The actomyosin cytoskeleton is therefore an essential component in force generation during this process.”

3) While the "seaming" is interesting, dorsal closure can occur without the seams. See Wells et al., 2014. The authors need to cite this paper and rationalize why they focus on the "seaming" process, which does not seem to be the major force generator.

Epidermal seaming is essential for the completion of dorsal closure. A suppression of seaming by microtubule depolymerization in *Drosophila* leads to incomplete closure (See Jankovics and Brunner, Dev. Cell. 2006). The work of Wells et al., 2014 describes how closure proceeds after ablation of the canthi. In this case, closure proceeds and seaming along the epidermis still takes place at other locations than the canthi. This is explicitly mentioned at the end of the first paragraph of the Results section of Wells et al., 2014:

“…Moreover, edge-to-edge connections were formed at a variety of different locations between the opposing parallel leading edges of the advancing lateral epidermal cell sheets […] Such edge-to-edge closure is in contrast to the majority of native closure, where seams are formed solely at the canthi during zipping. …Just as in native closure, a seam rich in F-actin formed upon the completion of closure and then disappeared as the epithelial sheet became continuous and seamless…”

Whether epidermal seaming generates forces contributing to dorsal closure is still unclear and an open question. However, a slowing down of closure is observed by Wells et al., 2014 towards the end of dorsal closure in canthi ablated embryos, suggesting that seaming is likely to generate forces contributing to closure.

We have modified the manuscript to discuss the influence of seaming: “Still an open question is whether microtubule-dependent epithelial seaming is a process that can generate forces. In the case of *D. melanogaster*, laserablation of epidermal canthi (*i.e.* the epidermal corners where opposing epidermal flanks meet) slows down dorsal closure (Wells et al., 2014); however, an F-actin-enriched epidermal seaming still occurs between the opposing leading edges of the epidermis despite canthi removal.”

[Editors' note: the author responses to the re-review follow.]

[…] Collectively, all three reviewers suggest that in addition to toning down the strength of the statements that imply mechanism, the authors must work on generating high quality data to strengthen the claims made about the cell biology of the process, taking into account several useful points made by the reviewers, for future submissions.

Reviewer #1:

In their revised manuscript, Solon and colleagues have attempted to address some of the queries raised by the reviewers in the previous version. Notably, they present some orthogonal views and additional images and analysis to better define tissue topology and perform new laser ablation experiments to test the requirement for the cellular behaviours (serosal ingression and the two seaming events) in the process. I appreciate the challenges and excitement of studying a new model organism, and that it is not trivial to obtain mechanistic cellular and molecular insights. I also appreciate the effort the authors have made to address the questions raised. I still feel that the description of the process and the cellular and subcellular mechanisms invoked as being important are not sufficiently strongly backed up by the data presented. Without a clear description of the topology of the process (even from fixed preparations with good orthogonal views), it is hard to interpret the consequences of the drugs used to perturb actomyosin contractility and the microtubule network, both of which appear to have widespread effects (including the patterning of and shape changes in the epidermis). Also, it is not possible to conclude much from the laser ablation experiments as presented here, which are very qualitative, other than that in some instances, it produced recoil and the process failed. I therefore feel that the statements made in the paper that relate to i) physical interactions between the participating tissues, ii) requirement of actomyosin contractility and iii) the two microtubule rearrangements for specific cell behaviours during dorsal closure are still too strong. I think the quantitative description of the tissue dynamics presented here (serosal area and height dynamics), if combined with a good cell biological description of the tissue topologies, interactions and cell behaviours and cytoskeletal organization, will in itself constitute an important piece of work in a new model organism and getting better quality data to strengthen these aspects will add significantly to our understanding of the conservation of developmental mechanisms.I list my detailed comments below:1) While it is clear from the videos that serosal rupture and contraction (reduction in area of the sheet) occur, it is still not clear to me what the authors mean by serosal ingression. Looking at the videos, it appears as if the serosa ruptures and retracts, collapses over the dorsal opening and then degenerates. While the schematics are useful, the data presented do not convincingly reveal the earlier close apposition of the serosa and the vitelline membrane, serosal ingression, or the subsequent reapposition of the serosa and vitelline membrane (as depicted in the schematics in Figure 1). What would be very useful, as suggested earlier, is a staged series of images of confocal projections in a lateral view, and dorsal-up views with their orthogonal sections. Phalloidin and DAPI should reveal these topologies and, at least in Drosophila, some phalloidin can get in through the vitelline membrane.

The reviewer is concerned about the topology and positioning of the serosa tissue over time and, particularly, its relation with the vitelline envelope. We agree with the reviewer that the serosa tissue ruptures, retracts and a part of serosal tissue accumulates over the dorsal opening and will degenerate. However, we observe that, concurrently, a part of serosal tissue bends towards the yolk granules with an apicobasal elongation and is internalized into the embryo. To improve the clarity of the manuscript, we replaced the word “ingression” by the word “internalization” to describe the inwards bending of the serosa and eventual seaming of the epidermis on top. We now provide in the Figure 1—figure supplement 3 and Figure 4—figure supplement 2 additional confocal imaging of live embryos injected with FM-4 64 showing these changes in topology and morphology of the serosa during retraction and internalization of this tissue and have modified the text of the manuscript to better describe this process. As the reviewer also points, the schematics in Figure 1 were misleading, particularly on the potential relation between the serosa and the vitelline envelope. We have modified the schematics to clarify this point and would like to point out that our work aims to describe the sequence of morphogenetic events occurring during dorsal closure in *Megaselia abdita* embryos rather than the precise anatomical relationship between the serosa and the vitelline envelope.

2) The authors measure the areas of serosal cells at different stages during dorsal closure and show that the apical areas reduce over time. This addresses a query I had raised in the previous version. The authors also show that the area of the serosa rapidly reduces. It would be desirable to see in an orthogonal view, what happens to the apicobasal axis of this sheet during contraction and ingression.

As proposed by the reviewer, we have added some orthogonal views of phalloidin-stained embryos showing an apico-basal reduction of the serosa cells during their retraction and internalization. This reduction correlates with an apical accumulation of F-actin. This is now represented in Figure 3—figure supplement 1 and is supported by the time-lapse sequence in FM4-64-labeled embryos in Figure 1—figure supplement 3 and Video 2 where an apico-basal elongation of the serosa cells is also observed during the retraction.

3) The correlation between cell elongation and microtubule organization suggests that the latter follows the former and may be a consequence of shape change (subsection “A microtubule-based seaming of the extraembryonic amnion is required for dorsal closure in Megaselia abdita embryos”, first and fourth paragraphs). The colcemid experiments indicate that the microtubule cytoskeleton is necessary for the timely completion of closure but do not point specifically to a role for microtubule rearrangements in the seaming events described.

Our experiments indicate that the microtubule cytoskeleton is necessary for the completion of dorsal closure and that the microtubules are aligned in both epidermis and amniotic tissue. This microtubule alignment correlates with cell elongation. We agree with the reviewer that our results do not point specifically to a role for this rearrangement in the seaming events. However, we show that similarly to *Drosophila* dorsal closure, microtubules in *Megaselia* embryos also align in tissues undergoing seaming (in this case amnion and epidermis) and are necessary for seaming, pointing towards a general mechanism for epithelial fusion. We have rephrased our results in the manuscript to tone correctly this message and to avoid overstatements.

4) The effects of both the ROCK inhibitor (subsection “A microtubule-based seaming of the extraembryonic amnion is required for dorsal closure in Megaselia abdita embryos”, second paragraph, Video 5) and colcemid (Video 10) appear to be widespread. Specifically they do not help in resolving causality between cytoskeletal organization and cell behaviour. Both also appear to alter cell shapes/ pattern in much of the epidermis, which may also contribute to the effects on dorsal closure.

The reviewer is concerned about the effect of our pharmacological treatment on the different tissues involved in dorsal closure. This is a difficult point to address due to the lack of genetic tools in this non-model organism that could allow targeting specific tissues. To overcome this, we have performed additional live imaging that focuses on cell shape changes in the serosa upon drug treatment. These additional experiments show that Rhok inhibition blocks the serosal shape changes (i.e.apico-basal elongation and size reduction) and serosa bending and internalization. In the case of colcemid treatment, serosal shape changes and tissue bending are not affected, only complete internalization of the serosa is perturbed, consistent with microtubule being essential for amnion seaming but not for serosal retraction and remodelling. In addition, we show that amnion cells in wild type and colcemid-treated embryos have a similar elongation during serosa retraction and dorsal accumulation. These data are now in Figure 3—figure supplement 2, Figure 4—figure supplement 2 and the new Figure 4—figure supplement 4, respectively. Finally, we have performed colcemid deactivation experiments by irradiating UV light in an area covered by the amnion cells prior to seaming in colcemid-injected embryos. These experiments are directed to reactivate microtubule polymerization, specifically in the amnion, and resume amnion seaming. Indeed, we observe a partial rescue of the process and progression of dorsal closure indicated by the reduction of the height (*h*) of the dorsal opening (see Figure 4—figure supplement 3). We believe these experiments indicate a role of the microtubule cytoskeleton in the seaming of the amnion during dorsal closure in *M. abdita*.

5) The statements made about attachment between the tissues (Figure 2) are too strong and the schematics are not backed up by the data in the images. It is probably okay to say close apposition but active attachment is not reflected by the data shown (subsection “Dorsal closure in Megaselia abdita involves synchronized serosa rupture and epidermal progression”, third paragraph).

We agree with the reviewer and have changed the wording of the manuscript to replace attachment by close apposition as the reviewer suggests.

6) In the laser ablation experiments, the ROI ablated should be marked, the power regimes used and the number of embryos ablated for each experiment should be mentioned in methods. While some experiments show tissue recoil, the effect of ablation on the completion of dorsal closure is less clear. Overall, these experiments appear to be very qualitative.

These experiments are challenging due to the low signal to noise ratio of the FM4 64 labeling and to potential contribution to the kinetics of a wound healing response. We have removed these experiments and, in complement, we have performed UV deactivation of colcemid. We believe colcemid-deactivation experiments are more informative on the requirement of microtubules in the amnion.

Reviewer #2:

The authors have comprehensively responded to the reviewers’ concerns. They added new stainings and analyzed cross section to clarify the tissue topology. They also conducted new functional experiments; in particular, they performed laser ablations of serosa and amnion cells followed by live imaging. Most of the new material is convincing and supports the main conclusion of the paper that two subsequent epithelial fusion events are required for dorsal closure in Megaselia. I have however still a serious problem with variable quality of figures and videos.Here is a list of figures and videos of low quality:Figure 1—figure supplement 1B, B´ I would remove this zen ISH. Both could be artefacts. This result is not important for the paper.D plus Video 1: it is not clear what one can learn from this video. I suggest removing.

We agree and have followed the reviewer’s suggestion. These data have been removed from the manuscript.

It is very laudable that the authors tried to do laser ablations of amnion and serosa cells. However, the video and stills from these ablations are very hard to interpret. The side of ablation and the cellular consequences are hardly visible. I am not able to interpret the pictures shown in Figure 4—figure supplement 3 and Video 8 does not help. The same applies for the amniotic ablations shown in Figure 4—figure supplement 5. Indeed, I found none of the laser ablation movies (Video 6–Video 8) informative.

As mentioned to the reviewer 1, the laser dissection experiments are very challenging on this non-model organism. We have replaced these experiments by UV deactivation of colcemid in an area covering the amnion during serosa internalization in colcemid-injected embryos. These experiments are directed to reactivate microtubule polymerization in the amnion and resume dorsal closure progression. Indeed, we observe a partial rescue of the process indicated by the reduction of the height (*h*) of the dorsal opening (see Figure 4—figure supplement 3). We find these experiments more informative on the impact of microtubule depletion in the amnion and the requirement of microtubules for dorsal closure progression.

It is also laudable that the author tried to produce optical cross section which in many cases helped to better understand tissue arrangements. However, the cross sections shown in Figure 4—figure supplement 6 seem not to be informative to me.

We have improved our live imaging acquisition parameters to obtain orthogonal views (cross sections) of embryos at the late stages of dorsal closure (during serosa internalization). These videos and still images of orthogonal views are included in Figure 1—figure supplement 3, Figure 3—figure supplement 2 and Figure 4—figure supplement 2. These images show the apico-basal elongation and internalization of serosa cells and the effects of the different drugs on these cell shape changes.

Taken together the paper has clearly improved. But the authors unfortunately added material which does not really show what they want to say. I think most of this material can be just removed.

We have followed the reviewer’s suggestions and, in one hand, removed low quality data from the manuscript, on the other hand, we have improved the data quality in other sections of the manuscript. We believe the new added data, particularly the cross sections kinetics in the different drug treatment, are of better quality and have significantly strengthened our previous observations.

Reviewer #4:

[…] While I find the manuscript conceptually intriguing, I am not sure that all major conclusions are sufficiently supported by the presented data. In addition, I have some issues with methods and data quality.Comments on major conclusions:1) […] The authors show that in zen RNAi embryos microtubules are no longer aligned as in wildtype (Figure 4—figure supplement 2'), yet dorsal closure still occurs in 1/3 of zen RNAi embryos. The authors suggest that dorsal closure in zen RNAi embryos is due to incomplete gene knockdown, the development of a remnant serosa, and a local alignment of microtubules close to the remnant serosa. The key observation here is an accumulation of local F-actin, which the authors interpret as remnants of serosal tissue [subsection “A microtubule-based seaming of the extraembryonic amnion is required for dorsal closure in Megaselia abdita embryos”, third paragraph]. I do not think that local actin accumulation is sufficient evidence to interpret cells as remnants of serosal tissue, in particular since main conclusions of the paper depend on this result (1. microtubules need serosa to align, 2. aligned microtubules are required for dorsal closure). To support their interpretation convincingly, I would suggest the authors build on their defined tissue-specific features of the serosa [subsection “Dorsal closure in Megaselia abdita involves synchronized serosa rupture and epidermal progression”, first paragraph], and use them to demonstrate the presence of serosal remnants in zen RNAi embryos by marker gene expression (zen, ddc could be an alternative (Rafiqi et al., 2010)), nuclear size, and DNA compactness.

The reviewer questions the implication of the serosa in the observed alignment of microtubules in the amnion. We would like to state that this is not a conclusion of the manuscript. This misunderstanding might have arisen due to our interpretation of the *zen*-RNAi results, thus, we would like to apologize for this.

The main conclusions of our manuscript are:

1) Dorsal closure in *M. abdita* (a three rather than two tissue system) involves the rupture and contractility of serosal tissue, 2) the roles of the *JNK*/Dpp signaling pathway and an actomyosin epidermal cable during dorsal closure are conserved in *M. abdita* compared to *D. melanogaster*, and 3) morphologic and kinetic analysis of dorsal closure reveals two consecutive epithelial seaming process (amniotic as well as epidermal) that depend on the microtubule cytoskeleton.

In the revised manuscript, we have removed the results of the *zen*-RNAi and laser ablation of the serosa since they are low quality data and can lead to misinterpretation. Instead, we have performed a rescue experiment on colcemid-injected embryos by deactivating the drug using UV irradiation in an area of the embryo corresponding to the amnion. As explained to reviewers #1 and #2, these experiments are directed to reactivate microtubule polymerization in the amnion and resume dorsal closure progression. Indeed, we observe a partial rescue of the process indicated by the reduction of the height (*h*) of the dorsal opening (see Figure 4—figure supplement 3). Our conclusion from these rescue experiments is that a polymerized microtubule cytoskeleton in the amnion is required for the progression of dorsal closure in *M. abdita* embryos.

2) The conclusion that the amnion fuses by seaming along the dorsal midline is based on time lapse recordings after FM4-64 injection (Video 9). The video shows how the flanking cells of the amnion meet along the dorsal midline after the serosa is internalized. Following serosa internalization and amnion touching, the ectoderm meets along the dorsal midline. The two processes seem to follow very different dynamics. In case of amnion cell meeting along the dorsal midline, cells touch and close a gap over the serosa at a speed of about 6 µm/min (i.e. 30 µm at 1:10-1:15). In case of the ectoderm cell meeting along the dorsal midline, cells touch and close a gap over the amnion at a speed of about 1.2 µm/min (i.e. 30 µm at 1:50-2:15). While the dynamics in the slow fusion in the ectoderm appear consistent with the idea of seaming, I am concerned that the fast touching of the amnion cells is not an equivalent process. Rather than a seam of fused cells, the somewhat brighter cell boundary between touching amnion cells along the dorsal midline may just reflect the increased intensity due to the touching membranes of two amnion cells and/or remnants of actomyosin accumulation at the serosa/amnion interface.

The reviewer is concerned about the differences in velocities in amniotic seaming and epidermal seaming, therefore questioning the potential similarities between the two processes. Seaming velocities are dependent on different factors such as the mechanical properties of the adhering tissues, their seaming angle or the contraction of the tissue covering the gap. In *Drosophila*, changes (up to a factor 2 in velocities) in epidermal seaming are observed during dorsal closure (Peralta et al., Biophys J, 2007). An estimation of amnion and epidermis seaming velocities in *M. abdita* embryos (8.1 ± 2 µm/min, SD, *n*= 5 embryos for the amnion and 3.8 ± 1.7 µm/min, SD, *n*= 5 embryos for the epidermis) indicate a variation of about a factor 2, which could also be due to differences in mechanics between amnion and epidermis. In the revised manuscript, we have included the estimation of the seaming speeds in the amnion and epidermis and discussed the differences in velocities between the two processes. These estimates can be found in the second paragraph of the subsection “A microtubule-based seaming of the extraembryonic amnion is required for dorsal closure in *Megaselia abdita* embryos”.

3) The conclusion that microtubule organization in the amnion is required for epithelial seaming and dorsal closure is based on time-lapse recordings after injection of colcemid, which leads to what is described as "dorsal closure arrests during the initial stages of serosa ingression". I am not quite sure the Video shows an "arrest". When serosa internalization does not proceed further (02:10), dorsal closure seems to reverse rather than being arrested. I am worried that instead of showing a specific effect of microtubule depolymerization, this is due to a collapse of the system and dying of the embryo. Because this is a key experiment aimed to demonstrate a role of microtubules in dorsal closure, it would be necessary to address such concerns by providing more information (how often has this phenomenon been observed in independent experiments, what are possible explanations for the observed serosa eruption) and/or by a more local perturbation of microtubule organization in the amnion.

The reviewer is concerned about the reproducibility and specificity of the phenotype observed with the microtubule depolymerizing drug colcemid. We point out that the graph in Figure 4 shows the average kinetics of 15 injected embryos and all of them show a transient arrest of closure before showing a collapse or “reverse” of the system. In addition, we have checked the effect of the drug on the microtubule cytoskeleton as observed in Figure 4—figure supplement 2. The observed “arrest” phenotype is consistent with published results in *Drosophila*, where dorsal closure is also arrested at epidermal seaming stages after colcemid injection (Jankovics and Brunner, Dev Cell, 2006).

We have now complemented these experiments with additional live imaging showing similar cell shape changes in the extraembryonic tissues (apico-basal and dorso-ventral elongation in serosa and amnion, respectively) of colcemid treated embryos compared to wild type embryos (Figure 4—figure supplement 2’ and Figure 4—figure supplement 4’’ and corresponding Video 7 and Video 9). These additional data show that shape changes of serosa and amnion during dorsal closure *in M. abdita* are not impaired in colcemid-injected embryos, rather, dorsal closure is transiently arrested during serosa internalization, fails to progress and “reverses”, consistent with amnion seaming failure. In addition, UV-deactivation of colcemid in the amnion region rescues partially the closure, indicating a specific role for microtubules in the amnion (Figure 4—figure supplement 3).

4) The conclusion that amnion seaming is a prerequisite for epidermal seaming is based on laser-ablation. Following laser ablation of what is described to be the amniotic-seaming front, epidermal seaming stops and results in a late dorsal open phenotype. It is difficult to follow the experiment based on the provided time-lapse recordings. In the wildtype reference (Video 9), ectoderm, amnion, and serosa cells can be clearly distinguished. In the time-lapse recording that is meant to show ablation of the amniotic-seaming front (Video 11), neither amnion nor serosa cell outlines are visible. Thus it is not possible to localize the laser ablation and what tissue has been affected. The visualization of the amnion, the area in which the amnion cells touch along the dorsal midline, and the ablation precisely where amnion cells touch, is, however, critical to demonstrate that dorsal closure is impaired due to interference with "amnion seaming" (see above for concerns about "seaming") and not just due to the loss of amnion tissue integrity.

As mentioned above, these data have been replaced by UV deactivation of colcemid in the amnion area. These additional experiments are consistent with a requirement of the microtubule cytoskeleton in the amnion for dorsal closure progression and support an amniotic seaming.

Comments on data quality:1) The authors use nuclear size and DAPI density to characterize tissue type [subsection “Dorsal closure in Megaselia abdita involves synchronized serosa rupture and epidermal progression”, first paragraph]. How was area and density of DNA stain in Megaselia defined as faithful descriptors of cell type? In its current presentation, the coloration of nuclei appears a bit arbitrary, even though it is most likely based on a double staining of marker genes (zen for serosa, pnr for amnion) and DNA stain to correlate nuclear size and DNA density with tissue types. To explain their rationale better, the authors should add an image of a such double staining and illustrate how the quantitative and tissue-specific parameters of nuclear size and "DNA compactness" were obtained.

The reviewer questions the anatomical description of cell type based on DNA staining for tissue type characterization. Nuclear anatomy and staining have been previously used to identify extraembryonic tissues in the flour beetle *Tribolium castaneum* (Panfilio et al., Biol Open, 2013). We extended this method quantitatively by measuring the staining profile, size of DAPI-labeled nuclei and their z position in 150 different cells from at least 15 embryos. A representative image of DAPI-stained cells is now shown in Figure 1—figure supplement 1. We used the pattern of DAPI staining to discriminate between serosa (discontinuous or “punctuated” staining) and amnion (continuous or “compact”) cells. Representative staining profiles for each cell type are shown in Figure 1—figure supplement 1’. The measured average size for each cell type are: 125 *±* 21 µm^2^ for serosa cells; 77 *±* 16 µm^2^ for amnion cells and 14 *±* 3 µm^2^ for epidermal cells. These measurements are plotted in Figure 1—figure supplement 1’’. In addition, spatial localization of extraembryonic cells was used as a descriptor. We observed that amnion cells (medium size nuclei with compact staining) form a 1-2 cell row located adjacent to the epidermis as previously described (Rafiqi et al., 2008), while serosa cells (largest nuclei with discontinuous staining) are homogenously distributed over the surface of the embryo as observed in Figure 1—figure supplement 1’. The different z-position of serosa and amnion cells was used to classify the cells. Serosa cells are more superficial than amnion cells and in some instances serosa cells can be observed on top of amnion cells, this can be observed in Figure 1’. We would also like to point out that our measurements of nuclear size and staining patterns are obtained from z-confocal images, thus, we know exactly at which z-position the nuclei are. Using these quantitative descriptors (staining profiles, nuclear size and z position) we pseudo-colored in red the nuclei of serosa cells (largest nuclei with discontinuous staining) and in blue the nuclei of amnion cells (medium size nuclei with compact staining).

We initially used in situhybridization (ISH) to stain for the marker genes *zen* (serosa) and *pnr* (amnion), however, as pointed out by reviewers #1 and #2, these stainings are of low quality and difficult to interpret. ISH stainings were removed from the manuscript as suggested by the reviewers. We believe that the quantitative description of nuclear anatomy between serosa and amnion cells, together with their localization in the embryo, allows us to discriminate between serosa and amnion cells. This quantitative description and the resulting measurements are now explained in more detail in the manuscript with additional images and graphs in the Figure 1—figure supplement 1.

2) Classification of serosa and amnion in Figure 1—figure supplement 1 is puzzling. The red nucleus on top of the third blue nucleus from the right seems to have a size that fits into the amnion category. What is the basis to classify this nucleus as serosal? Is it the density of the DNA stain? It does not become clear by which procedure two qualitatively different parameters (nuclear size and "DNA compactness") were compared if they contradict each other in the classification of a particular nucleus. Probably this is done by some sort of quantitative measure – the authors should comment on this procedure in more detail in the Materials and methods section.

As mentioned above, a combination of anatomical parameters was used to classify the nuclei into serosa or amnion. These parameters include DAPIstaining profile (continuous vs. compact), the size of the nucleus (125 *±* 21 µm^2^ vs. 77 *±* 16 µm^2^), the z-position of the nuclei (superficial vs. less superficial) and the spatial localization of the nucleus in the embryo (homogenously distributed in the embryo vs. a 1-2 cell row adjacent to the epidermis). In the specific case of Figure 1—figure supplement 1, we agree with the reviewer that this image did not reflect the methodology used to classify the nuclei. We have replaced this figure with a typical DAPI-stained embryo before and after extraembryonic nuclei identification (Figure 1—figure supplement 1’) and a close-up view of DAPIstained extraembryonic serosa and amnion nuclei and embryonic epidermis, together with the representative staining profile for serosa and amnion cells and the graph of nuclear size measurements (see Figure 1—figure supplement 1’’). This description is now explained with more detail in the Materials and methods section.

3) The density of serosa cells in Figure 1—figure supplement 1 are unexpectedly variable, it seems to be much higher in areas where the serosa covers the ectoderm than in areas covering the interface of amnion and ectoderm, the amnion, or the yolk. In previous publications and other images within this publication (e.g. Figure 1, Figure 1—figure supplement 2), the density of serosa nuclei appears to be rather even along the dorsal to ventral circumference of the embryo. Could the authors offer an explanation for this discrepancy?

We agree with the reviewer in that the density of serosa cells in Figure 1—figure supplement 1 seems uneven. The initial purpose of Figure 1—figure supplement 1 was to illustrate the difference in nuclear sizes among the three tissue types, for this reason we did not notice a particular uneven distribution of the serosa nuclei in this image, we apologize for this. The uneven distribution of the serosa cells in this image could be due to the fixing treatment affecting the geometry of this particular embryo or simply to inhomogeneities at the particular spatial resolution.

We point out that a similar close-up view on Figure 1 could as well result in an apparent inhomogeneous distribution of serosa cells. To avoid misinterpretations we have replaced this image by a representative view of the whole embryo showing the distribution of DAPI-stained serosa amnion and epidermal cells in the new Figure 1—figure supplement 1’ and a representative close-up view of a similar DAPI staining to show the difference in nuclear size in the serosa, amnion and epidermis in Figure 1—figure supplement 1’’.

4) Whole mount in situs of Mab_zen and Mab_pnr are not of high quality. In situs in Megaselia should be an established procedure, also for embryos at late stages (e.g. Rafiqi et al., 2010). In their description of gene expressions, the authors do not mention the staining of Mab_zen and Mab_pnr probe in the head and Mab_zen probe along the ventral midline. Do they have reasons not to trust this aspect of the staining? Dorsal staining of Mab_zen probe and Mab_pnr probe could be genuine, but this is difficult to evaluate at this quality level: poorly devitellinized embryos can lead to similar dorsal staining through unspecific probe accumulation at cracks in the vitelline membrane.

We agree with the reviewer that the staining in the whole mount in situ hybridization (ISH) presented in the manuscript is not very sharp. We think this could be due to the low expression levels of *Mab-zen* and *Mab-pnr* at these particular stages. ISHs at this late stage of development (dorsal closure) have not yet been reported. The latest stage of ISH in *Megaselia* embryos published in (Rafiqi, et al., 2010) is at germband retraction (stage 12, ~ 8 hours after egg laying), while dorsal closure occurs at stage 15 (~ 14-17 hours after egg laying, see Wotton et al., 2014). Recently published ISHs of *Megaselia* embryos at dorsal closure stage (Kwan et al., *eLife*, 2016) show that the expression levels of the gene *Mab-egr* in the amnion decreases towards stage 15 (see Figure 2—figure supplement 3I in Kwan’s paper). Similar decrease could also occur for *Mab-pnr* and *Mab-zen* at dorsal closure stage.

In any case, we would like to stress that the *Mab-zen/Mab-pnr* ISHs are a minor point of the manuscript. As suggested by the reviewer 2, we believe these data are not bringing significant information to the manuscript and have not been included in the revised manuscript.

5) The authors suggest that devitellinization results in damage of the dorsal region of the embryo [subsection “Dorsal closure in Megaselia abdita involves synchronized serosa rupture and epidermal progression”, third paragraph], where they do not detect phalloidin staining (Figure 1—figure supplement 2). This suggest the authors have evidence that the amnion is normally closed at this point of development. This is contradicting the schematic overview in Figure 1'. To clarify, it would be helpful to provide a brief description and maybe a panel with a closed amnion to illustrate the progression of wildtype amnion development.

The argument of the reviewer on how our results suggest that we have evidence for a closed amnion during dorsal closure in *M. abdita* appears unclear to us. We think that our observation of a region on the dorsal side devoid of cells after vitelline envelope removal, rather points towards a model in which only serosa cells covers this gap. This is supported by the fact that we always observe a homogeneous distribution of nuclear sizes attached to the removed vitelline membrane and do not observe amniotic cells on the most dorsal part of the envelope, either in fixed embryos (Figure 1—figure supplement 2) or in live DAPIstained embryos (Figure 1—figure supplement 1’’).

We point out that our morphological description of extraembryonic tissues at dorsal closure stage indicates that the amnion is a 1-2 cell row localized adjacent to the epidermis. This is consistent with previous description of the amnion (Rafiqi et al., 2008) and published ISH staining (Kwan et al., *eLife*, 2016). With live imaging, we observe that the amnion cells form a row apposed to the serosa cells and are absent on the dorsal part on top of yolk granules (Figure 1—figure supplement 2’’ and Figure 1—figure supplement 1’’). We also observe that the opposing single rows of amnion cells adjacent to the epidermis undergo elongation upon serosa internalization (Figure 4—figure supplement 4’’ and Video 9) and that these opposing amniotic flanks merge at the dorsal midline after serosa internalization (Figure 3’’, Figure 4’’ and Figure 4—figure supplement 1’). Altogether, these data point towards a model of an “open” configuration of the amnion at dorsal closure stage in which these cells are a 1-2 cell row adjacent to the epidermis and apposed to the serosa cells as depicted in our schematic overview in Figure 1’ and G-G’’ and in Figure 4’.

6) What is the stage of dorsal closure in Figure 1—figure supplement 2'? In the schematic overview (Figure 1—figure supplement 2) it seems as if the serosa still envelopes most part of the embryo. The embryo shown in 2D-D' seems to be older and serosa internalization well under way, corresponding to the schematic overview in Figure 1'. The authors explain that serosa tissue can be lost during devitellinization, suggesting that they used alternative landmarks to stage dorsal closure progression. It would be helpful if they could comment on their use of landmarks and how the embryo in 2D-D' was staged.

The embryo shown in the old Figure 1—figure supplement 2’ is at an early stage of dorsal closure, prior to serosa rupture. We apologize for not mentioning the landmarks used to stage this specific embryo. The general staging landmarks were mentioned before in the manuscript and include: a) the fusion of the dorsal ridge at the most anterior part of the embryo prior to serosa rupture (see time of dorsal ridge fusion depicted by a pink bar in Figure 1 and red curve depicting serosa area covering the embryo, as well as red bar landmarks in Figure 1—figure supplement 3’’); and b) the height (*h*) of the dorsal opening (see blue curve in Figure 1 and yellow lines and blue bar in Figure 1—figure supplement 3’’). In the case of the old Figure 1—figure supplement 2’ (new Figure 1—figure supplement 2’’), the dorsal ridge is fused but the height of the opening is still very large, indicating that the embryo is within 30 min after fusion of the dorsal ridge (following the graph in Figure 1). Also, older embryos in which serosa internalization is well under way, show an increased phalloidin staining due to F-actin accumulation at the most dorsal part of the embryo as observed in Figure 3 and Figure 3—figure supplement 1. As suggested by reviewer #1, we removed the old schematics of Figure 1—figure supplement 2 since our results do not show an amnion-serosa attachment, but rather indicate an apposition. As a result, we have modified the Figure 1—figure supplement 1 and the mentioned image now corresponds to Figure 1—figure supplement 2’’ and the staging landmarks are mentioned more explicitly in the figure legend of the revised manuscript.

7) The authors suggest that amnion and serosa share a junction or attachment site [subsection “Dorsal closure in Megaselia abdita involves synchronized serosa rupture and epidermal progression”, third paragraph]. The increase of phalloidin staining in Figure 1—figure supplement 2' is sufficient evidence that the cell with blue arrow touches the cell with white arrow. A blob of actin does not provide evidence for junctions, and the black lines indicating amnion-serosa attachment in schematic in Figure 1—figure supplement 2 may be interpreting the available data a bit too much.

We agree with the reviewer and apologize for our overinterpretation of the data regarding amnion and serosa apposition. As mentioned above, we have modified the figures and text in the manuscript to replace serosa-amnion attachment to serosa-amnion apposition.

[Editors' note: the final round of author responses follow.]

The manuscript has been improved but there are some remaining issues that need to be addressed before acceptance, as outlined below:Both reviewers agree that the revised version is a substantial improvement over the earlier versions both with respect to the conclusions and the readability. Specifically, the new data presented clarifies many issues on the topology of interacting tissues and the cell shape changes accompanying serosal contraction and provides a better resolution of the two seaming events. The reviewers were also impressed with the experiments designed to locally inactivate colcemid. Both reviewers had a few concerns that must be addressed (in a revision) and made the following suggestions.i) While the orthogonal sections presented do a good job of showing tissue topology, serosal ingression and its accompanying cell shape changes, the final configuration of the three tissues is not clear. It should be possible to resolve this by making orthogonal optical sections at different locations along the AP axis (at the canthus and closer to the AP midline) from the images presented in Figure 1—figure supplement 1'", phalloidin DAPI and Figure 4—figure supplement 1"', tubulin, DAPI, since closure proceeds inwards from the canthi.

We have followed the reviewers’ advice and extracted optical sections (through confocal stack re-slicing) along the anterio-posterior (AP) axis (sagittal sections) and orthogonal sections at anterior, mid and posterior positions perpendicular to the AP axis (cross sections). These optical sections were obtained from images presented in the previous Figure 3—figure supplement 1’’’ (phalloidin-DAPI stainings) and represent different stages of dorsal closure from serosa dorsal accumulation to epidermal seaming. These sections are now shown in a new Figure 1—figure supplement 4. We can observe through sagittal sections that serosa cells undergo internalization in the mid region of the dorsal opening (black arrowheads in a, b and c) and on the anterior and posterior sections, we can observe layers of cells under the superficial amnion layer corresponding to the internalizing serosa (c, c’, c’’’). Once the serosa cells have been fully internalized and the amniotic flanks have fused, the opposing epidermal leading edges move towards the dorsal midline and undergo epidermal seaming. Both the sagittal section (d) and the cross section at the mid position on the dorsal midline (d’’) show an even layer of epidermal cells followed by an uneven layer of extraembryonic cells underneath, indicating that extraembryonic cells do not accumulate under the closed epidermis (compared to the previous stage shown in c and c’’) and are rather removed, probably, in a similar fashion to the amnioserosa in *Drosophila* dorsal closure.

ii) The schematics depicting the changes in the three tissues participating in dorsal closure must be modified to accurately depict the serosal cell shape changes described and to indicate the final configurations of the three tissues at the end of the closure.

We have modified the schematics depicting the cellular changes in both the internalizing serosa and amnion cells during dorsal closure to accurately depict our experimental observations. We added a new Figure 4—figure supplement

5 to show the schematics in one sequence from the initial three-tissue configuration to the end of dorsal closure when epidermal seaming occurs and the extraembryonic amnion and serosa internalize.

iii) The data on the local colcemid inactivation includes two images and quantification of dorsal closure height. The authors must state when colcemid was injected and when it was inactivated and describe in Materials and methods how the ROI was created. It would be desirable to include the video of the treatment.

We included the information pointed out by the reviewers in the Materials and methods section of the manuscript. In summary, colcemid was injected during dorsal ridge fusion and prior to serosa rupture. Embryos were injected 30 min before imaging and UV-irradiation was performed during serosa dorsal accumulation and the initiation of internalization (i.e.approximately 20 min after serosal rupture). The ROI for UV-irradiation was created using the freehand selection tool in Leica software. We have now included the video associated to Figure 4—figure supplement 4 (new Video 10), where we can observe the rescue effect of UV-irradiation on a colcemid-injected embryo compared to a “non-irradiated” colcemid-injected embryo.

iv) Could the authors comment on the fate of the amnion: What happens upon fusion: does it detach from the epidermis? Or does it ingress with the serosa? Again orthogonal optical sections from the fused regions should help resolve this.

Upon fusion, amnion cells also internalize following serosa cells. Our recently added optical sections (Figure 1—figure supplement 4) show, at the end of dorsal closure, a layer of uneven extraembryonic cells underneath the continuous epidermis (Figure 1—figure supplement 4’, d’’ and d’’’). After internalization, amnion cells are probably removed by hemocytes in a similar way as the amnioserosa in *Drosophila* embryos. We have now included a sentence to discuss the internalization of the amnion cells in the manuscript:

“Optical re-slices of confocal projections along the anterio-posterior axis of fixed embryos indicate that extraembryonic tissues internalizes during dorsal closure and do not accumulate underneath the fused epidermis after completion of the process (Figure 1—figure supplement 4).”

v) Can the authors clarify if they are suggesting that the "movement " of amnion cells (devoid of actomyosin) is microtubule dependent and whether serosal contraction contributes to bringing the amnion flanks into close proximity?

We observe that the elongation of amnion cells also occurs in colcemid- injected embryos (see Video 9) and our data suggest that colcemid treatment affects essentially the seaming of amniotic flanks. Serosa contraction contributes to bringing the amniotic flanks into close proximity since Rho- kinase inhibition prevents serosa contraction and amnion cells do not seem to elongate (see Figure 3—figure supplement 2 and Video 4). We have modified the text in the manuscript to clarify this point:

“We observe that microtubule polymerization does not occur in colcemid-injected embryos (blue and white arrowheads for amnion and epidermal cells, respectively, in Figure 4—figure supplement 2 compared to 2D’) and that amnion cells initially elongate towards the dorsal midline as in wild type conditions, although they relax and retract from the amnion merging site (Figure 4—figure supplement 4 and Video 9). […] Our results indicate that serosa contraction contributes to bringing amniotic flanks into close proximity and that amniotic seaming is a microtubule-dependent event”.